# A nascent polypeptide sequence modulates DnaA translation elongation in response to nutrient availability

Michele Felletti[1], Cédric Romilly[2], E Gerhart H Wagner[2], Kristina Jonas[1]*

[1]Science for Life Laboratory and Department of Molecular Biosciences, The Wenner-Gren Institute, Stockholm University, Stockholm, Sweden; [2]Department of Cell and Molecular Biology, Uppsala University, Uppsala, Sweden

**Abstract** The ability to regulate DNA replication initiation in response to changing nutrient conditions is an important feature of most cell types. In bacteria, DNA replication is triggered by the initiator protein DnaA, which has long been suggested to respond to nutritional changes; nevertheless, the underlying mechanisms remain poorly understood. Here, we report a novel mechanism that adjusts DnaA synthesis in response to nutrient availability in *Caulobacter crescentus*. By performing a detailed biochemical and genetic analysis of the *dnaA* mRNA, we identified a sequence downstream of the *dnaA* start codon that inhibits DnaA translation elongation upon carbon exhaustion. Our data show that the corresponding peptide sequence, but not the mRNA secondary structure or the codon choice, is critical for this response, suggesting that specific amino acids in the growing DnaA nascent chain tune translational efficiency. Our study provides new insights into DnaA regulation and highlights the importance of translation elongation as a regulatory target. We propose that translation regulation by nascent chain sequences, like the one described, might constitute a general strategy for modulating the synthesis rate of specific proteins under changing conditions.

**\*For correspondence:**
kristina.jonas@su.se

**Competing interest:** The authors declare that no competing interests exist.

## Introduction

Most cells must be able to integrate external information with proliferative functions such as DNA replication, cell division, and protein synthesis. In particular, unicellular bacteria that frequently face drastic environmental changes in their natural habitat are able to precisely adjust their growth and division rate in response to environmental cues. While most bacteria proliferate rapidly under optimal conditions, they slow down growth and often enter a non-growing state under adverse conditions, for example, nutrient limitation (*Heinrich et al., 2015*). This dynamic switching between proliferative and non-growing modes contributes to stress and antibiotic tolerance, as well as the virulence of pathogenic bacteria (*De Bolle et al., 2015*; *Hobbs and Boraston, 2019*; *Wood et al., 2013*). Previous research has focused on the mechanisms that allow bacteria to globally reprogram gene transcription in response to nutrient availability (*Schellhorn, 2020*; *Steinchen et al., 2020*). More recently, the advance of ribosome profiling technologies has also revealed a central role of translational regulation in the adaptation to changing nutrient conditions (*Li et al., 2018*; *Parker et al., 2020*; *Subramaniam et al., 2014*; *Tollerson, 2020*). Despite significant progress, the precise molecular mechanisms regulating the translation of specific proteins during nutrient limitation are incompletely understood.

When entering growth arrest, most bacteria inhibit DNA replication initiation and halt their cell division cycle with a reduced number of fully replicated chromosomes (*Jonas, 2014*). Shutting down DNA replication initiation prior to the cessation of key metabolic functions and cell division likely preserves genome and cellular integrity. In nearly all bacteria, replication initiation depends on DnaA, an ATP-binding protein consisting of four distinct structural and functional domains: (1) a helicase

loading domain, (2) a linker domain, (3) an AAA+ (ATPase Associated with diverse cellular Activities) domain, and (4) a DNA-binding domain (*Hansen and Atlung, 2018*). DnaA is only active in the ATP-bound state and, upon binding to specific DnaA-boxes in the bacterial origin of replication, it unwinds double-stranded DNA and recruits the replisome. ATP hydrolysis following replication initiation subsequently inactivates DnaA (*Camara et al., 2005*; *Katayama et al., 2010*). Due to its critical function in replication initiation, DnaA has been proposed to be a crucial player in the nutritional control of DNA replication for more than 50 years (*Donachie, 1968*). Yet, the precise molecular mechanisms that regulate this important protein in response to changing nutrient conditions remain poorly understood.

In the freshwater bacterium *Caulobacter crescentus* (*Caulobacter* hereafter), DnaA levels decrease at the onset of carbon starvation when growth rate declines, which entails a block of DNA replication initiation (*Gorbatyuk and Marczynski, 2005*; *Lesley and Shapiro, 2008*; *Leslie et al., 2015*). *Caulobacter* is an important model for bacterial cell cycle studies due to its asymmetric cell cycle and its genetical tractability (*Govers and Jacobs-Wagner, 2020*). Furthermore, as a member of the *Alphaproteobacteria*, it is representative of a diverse group of bacteria, including important pathogens and agriculturally relevant species (*Govers and Jacobs-Wagner, 2020*). In its natural habitat, *Caulobacter* frequently experiences fluctuations in nutrient availability, for example, as an indirect consequence of seasonality or changes in the microbial composition of its niche (*Heinrich et al., 2019*).

*Caulobacter* DnaA is a relatively unstable protein that is degraded by the AAA+ protease Lon (*Felletti et al., 2019*; *Jonas et al., 2013*; *Liu et al., 2016*). Although Lon-dependent proteolysis is required for rapid clearance of DnaA at the onset of starvation, the rate of proteolysis is not significantly affected by changing nutrient conditions (*Leslie et al., 2015*). Instead, a starvation-induced decrease in DnaA translation rate was shown to be responsible for nutritional regulation of DnaA levels (*Leslie et al., 2015*). However, the underlying molecular mechanism remains elusive.

Here, we reveal the mechanistic basis underlying the starvation-induced downregulation of DnaA translation in *Caulobacter*. An in-depth biochemical and genetic characterisation of the *dnaA* mRNA showed that its 5' untranslated leader region (5'UTR) is dispensable for this regulation. Instead, inhibition of DnaA translation occurs during polypeptide chain elongation in a starvation-dependent manner, within a coding sequence downstream of the start codon. Our data are consistent with a model in which specific amino acids in the N-terminus of DnaA mediate a translation arrest in response to nutrient limitation while, or after, being added to the growing nascent polypeptide chain. In addition to providing new molecular insights into the nutritional regulation of DnaA, our findings illustrate how specific sequence motifs in nascent polypeptides can tune the first stages of translation elongation under changing nutrient conditions.

## Results

### The 5'UTR of the *Caulobacter dnaA* mRNA adopts a complex secondary structure

In *Caulobacter*, DnaA levels decrease when cells are shifted from a glucose-supplemented minimal medium to the glucose-limiting medium M2G$_{1/10}$ (*Figure 1A*; *Gorbatyuk and Marczynski, 2005*; *Leslie et al., 2015*). Previous work established that this carbon starvation-induced downregulation of DnaA abundance is caused by decreased *dnaA* translation in combination with constitutive DnaA degradation by Lon (*Leslie et al., 2015*). To elucidate the mechanistic basis for starvation-dependent downregulation of DnaA, we structurally and functionally characterised the mRNA region comprising the 155 nucleotides (nt) long 5'UTR and the first 118 nt of the *dnaA* coding region (total RNA length 273 nt) (*Schrader et al., 2014*; *Zhou et al., 2015*; *Zweiger and Shapiro, 1994*). Native polyacrylamide gel electrophoretic analysis of this in vitro transcribed $^{32}$P-5'-end-labelled RNA indicated a dominant conformation of the *dnaA* mRNA leader (*Figure 1—figure supplement 1A*). To determine its secondary structure, we combined mFold and ViennaRNA computational predictions with structural probing (*Figure 1—figure supplement 1B–D*). The latter was performed with Pb$^{2+}$ to induce RNA cleavages in single-stranded regions, RNase T1 to cleave after single-stranded G residues, and RNase V1 for mapping double-stranded stretches. The experimental and in silico analyses suggested seven base-paired segments: one helical element (P1) and six stem-loop motifs (P2–P7, *Figure 1B*). The 5'-most 8 nt are part of helix P1, which is connected via a three-way junction to stem-loops P2 and P3. The stable GC-rich stem P4 carries a G-rich loop which is highly susceptible to T1 RNase cleavage.

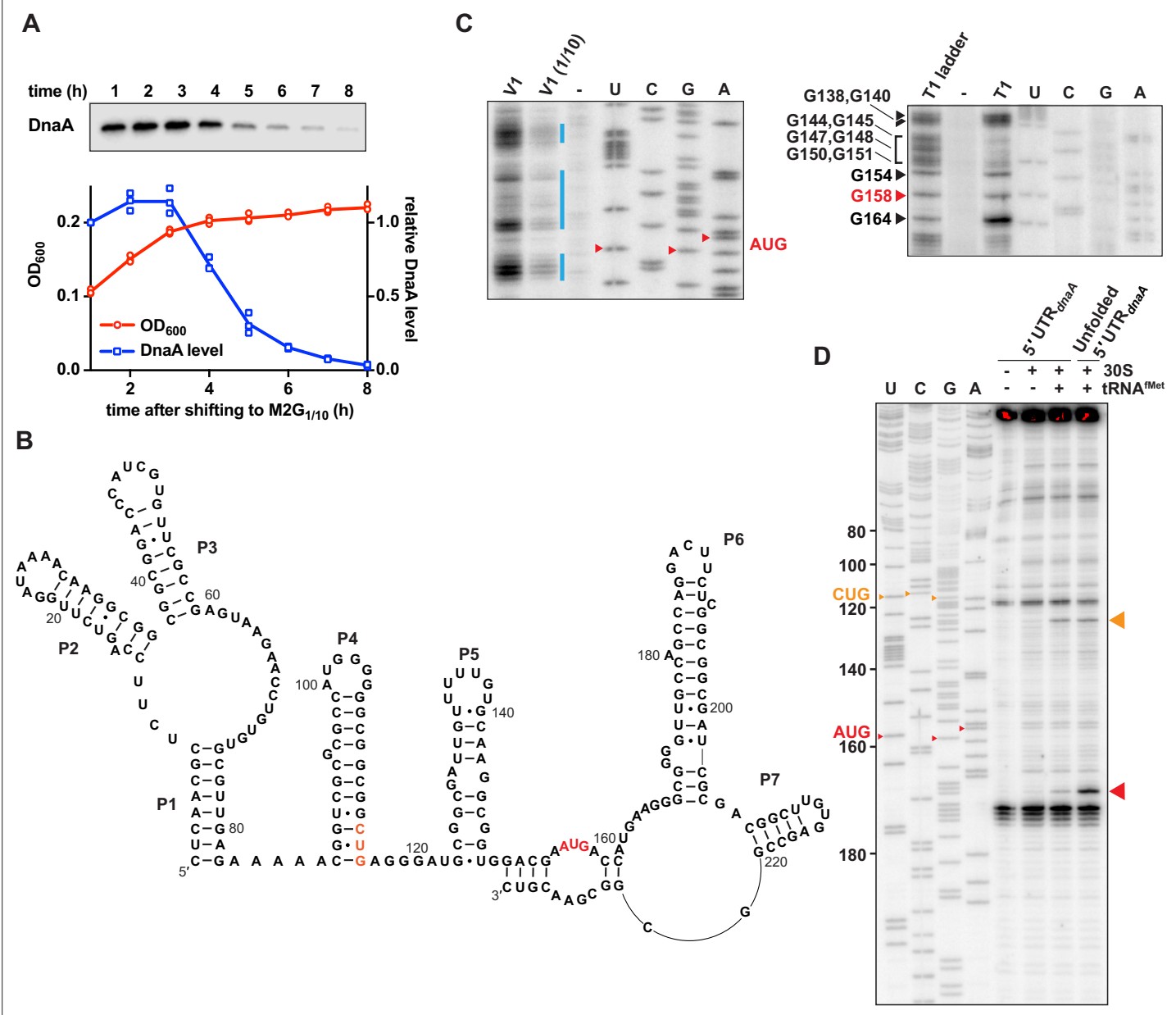

**Figure 1.** *Caulobacter dnaA* mRNA assumes a complex secondary structure that inhibits pre-initiation complex formation. (**A**) Growth curve (red) and DnaA abundance (blue) measured during a carbon exhaustion experiment. Wild-type *Caulobacter* was grown in a defined minimal medium (M2G$_{1/10}$) containing 0.02% glucose as the sole carbon source. DnaA levels were determined by western blot and band intensities quantified relative to the first time point (t = 1 hr). Averages of three independent replicates are shown with error bars representing standard errors. (**B**) Secondary structure of the mRNA region encompassing the 5'UTR and the first 26 codons of the *dnaA* open reading frame. The reported AUG start codon and the alternative CUG codon are highlighted in red and orange, respectively. See **Figure 1—figure supplement 2A** for an alternative secondary structure differing in the way the 3'-terminal RNA portion folds back on the start codon region. (**C**) RNA probing gels showing the *dnaA* mRNA region containing the translation start site and the RBS. Upon partially digesting a synthetic in vitro transcribed RNA with V1 (left gel) or T1 (right gel) RNases, primer extension ($^{32}$P-5'-end-labelled primer B; **Figure 1—figure supplement 2A**) was used to detect cleavage positions. V1 and V1 (1/10) lanes: samples treated with different concentrations of RNase V1; T1: sample treated with T1 RNase; T1 ladder lane: RNase T1 treatment under denaturing conditions; -: mock-treated RNA; U, C, G, A lanes: sequencing reactions using the $^{32}$P-5'-end-labelled primer B. The bands corresponding to the start codon nucleotides are indicated in red. The blue lines indicate the dsRNA regions. Complete gels are shown in **Figure 1—figure supplement 1C and D**. (**D**) Toeprinting assay showing the sites of pre-initiation complex formation in the *dnaA* 5'UTR. The reverse transcription step was performed using $^{32}$P-5'-end-labelled primer A (**Figure 1—figure supplement 2A**). Inclusion of *Escherichia coli* 30S ribosomes and tRNA$^{fMet}$ are indicated on the top of the gel. Unfolded 5'UTR lane, assay performed using a heat-unfolded in vitro transcribed RNA. U, C, G, A lanes: sequencing reactions using the $^{32}$P-5'-end-labelled primer A. The large red and orange triangles indicate the reverse transcriptase stops induced by the 30S initiation complexes formed at the AUG and CUG codons,

*Figure 1 continued on next page*

*Figure 1 continued*

respectively. The AUG and CUG codons are indicated on the sequencing lanes with small red and orange triangles, respectively.

The online version of this article includes the following figure supplement(s) for figure 1:

**Source data 1.** Source data for *Figure 1*.

**Figure supplement 1.** Structural characterisation of the *dnaA* mRNA leader.

**Figure supplement 1—source data 1.** Source data for *Figure 1—figure supplement 1*.

**Figure supplement 2.** Structure conservation of the *dnaA* mRNA leader and predicted interaction with *Caulobacter*'s 16S  rRNA.

**Figure supplement 2—source data 1.** Source data for *Figure 1—figure supplement 2*.

A fifth stem-loop structure (P5) is located upstream of the AUG start codon. Two mismatches in this stem confer some instability ($\Delta G_{folding}$ of –7 kcal/mol). The region around the start codon appears mostly unstructured, suggested by T1 cleavages at nucleotides G154, G158, and G164 (*Figure 1C*). The mRNA region encompassing the first 118 nt of the *dnaA* coding region folds into two stable stem-loops, P6 and P7. The additional RNase V1 cleavages in this portion of the RNA might indicate base-pairing of the 3' terminal nucleotides with the mRNA region around the start codon.

We also assessed the level of sequence conservation by BLAST using the 5'-most 233 nt of the *dnaA* mRNA of *C. crescentus* NA1000 as a query. Closely related sequences were found in several species of the *Caulobacter* and *Phenylobacter* genera (family *Caulobacteraceae*). Alignment of these 66 sequences revealed three conserved sequence domains. Further analysis, using the co-variance-based software CMfinder, identified three conserved independent structural domains corresponding to helices P1–3, P4, and P6 (*Figure 1—figure supplement 2B*), thus additionally supporting the probing results.

## The structure of the *dnaA* 5'UTR inhibits pre-initiation complex formation in vitro

One of the most striking features of the proposed *dnaA* mRNA structure is stem-loop P5. Its close proximity to the translation start site might suggest reduced access of initiating ribosomes. Since the *Caulobacter dnaA* 5'UTR had been reported to lack an obvious Shine–Dalgarno (SD) sequence (*Schrader et al., 2014*), we first predicted plausible sites of interaction with the 30S  ribosomal subunit. The base-pairing energy was calculated for an 8 nt sliding window of the 5'UTR and the anti-SD sequence of *Caulobacter*'s 16S  rRNA (*Figure 1—figure supplement 2C*), which suggested possible SD-like interactions between A143 and G150.

Experimentally, we assayed translation pre-initiation complex formation in vitro by toeprint analysis on the corresponding *dnaA* mRNA using purified *Escherichia coli* 30S  subunit and tRNA[fMet] (*Figure 1D*). We obtained a weak reverse transcription signal consistent with 30S-tRNA[fMet] binding at the reported AUG start codon, confirming that a pre-initiation complex is formed at the predicted SD (between position 143 and 150) that is buried in stem P5. Conducting the toeprint after heat-induced denaturation of the RNA gave a significantly enhanced signal. This shows that destabilisation of the proximal elements (i.e., P5, P6) overcomes the partial inhibition seen in *Figure 1D* and suggests that the mRNA adopts a conformation that limits pre-initiation complex formation.

It is noteworthy that our toeprint assay indicated a second band, consistent with complex formation at a non-canonical CUG start codon (nt 114–116), with an upstream SD which was also predicted in silico (*Figure 1D* and *Figure 1—figure supplement 2C*). However, our genetic experiments, as described in detail below, showed that this site is not used for translation initiation in vivo.

## A fluorescent reporter system to monitor post-transcriptional regulation of *dnaA* during carbon starvation

To study the *cis*-regulatory elements in *dnaA* mRNA that are involved in the post-transcriptional regulation of DnaA under carbon starvation, we developed an in vivo fluorescent reporter system. We translationally fused the *dnaA* promoter and 233 bp of *dnaA* (comprising the 5'UTR and the first 26 codons encoding the DnaA N-terminus, 5'UTR-N$_t$ hereafter) to the enhanced green fluorescent protein (eGFP) gene on a low-copy plasmid and transformed the construct into wild-type *Caulobacter* (*Figure 2A*). When shifting the resulting reporter strain from glucose-supplemented to glucose-limiting

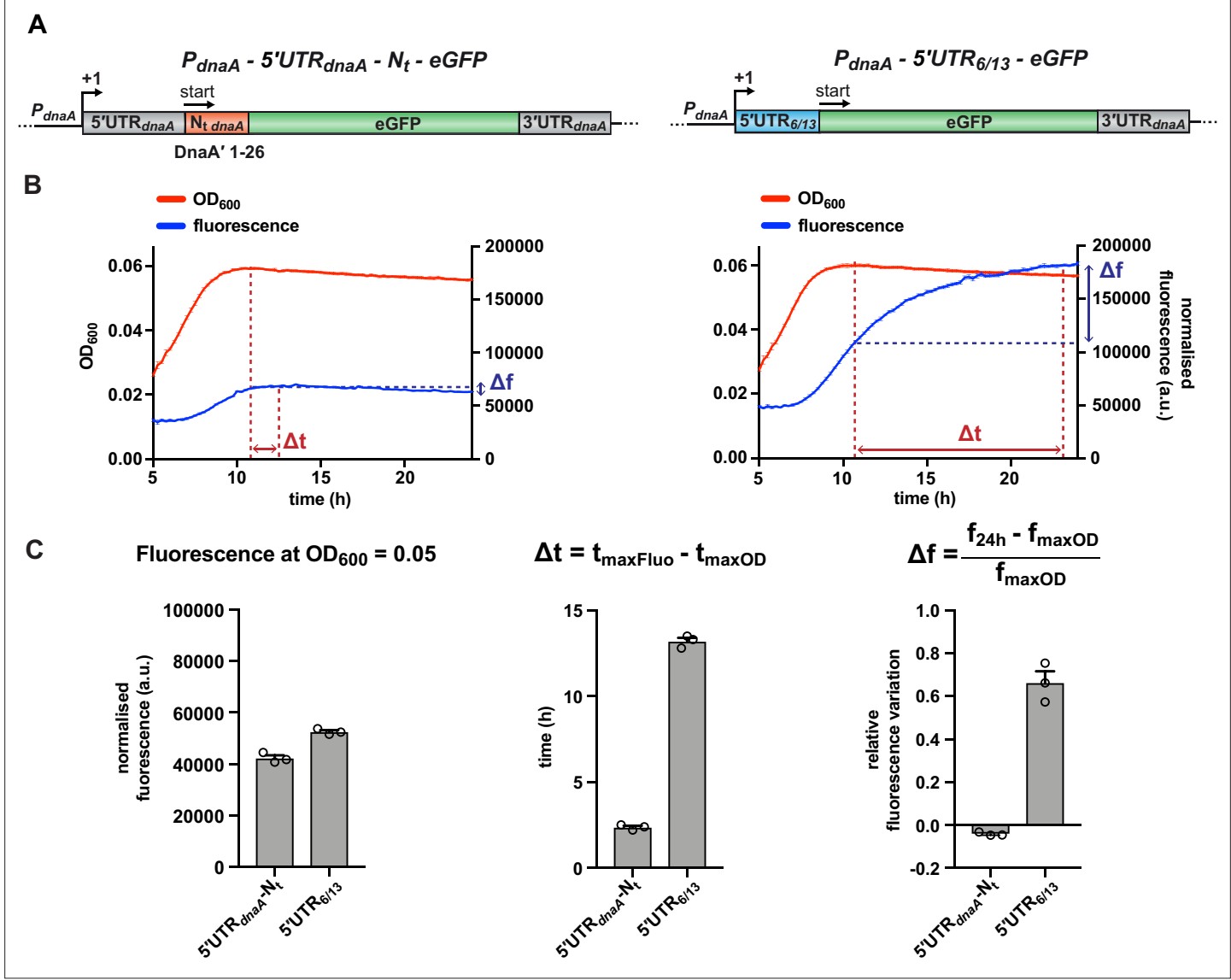

**Figure 2.** A fluorescent reporter system monitors the post-transcriptional regulation of *dnaA* during carbon starvation. (**A**) Schematic illustration of the plasmid-borne reporter constructs. In the 5'UTR-$N_t$ construct (left-hand side), the *dnaA* promoter ($P_{dnaA}$), the *dnaA* 5'UTR (5'UTR$_{dnaA}$; grey), and the first 26 codons of *dnaA* open reading frame ($N_{t\,dnaA}$; orange) were translationally fused to the eGFP gene (green), followed by *dnaA* intrinsic terminator (3'UTR$_{dnaA}$; grey). In the 5'UTR$_{6/13}$ construct (right-hand side), the 5'UTR$_{dnaA}$-$N_{t\,dnaA}$ module was substituted with the non-nutritionally regulated 5'UTR$_{6/13}$ (light blue). (**B**) Culture fluorescence was measured to monitor eGFP synthesis during growth in M2G$_{1/10}$ for 24 hr using a microplate reader. Background correction was performed at each time point by subtracting the fluorescence of a strain carrying a pMR10 empty plasmid (pMR10-BG, ***Supplementary file 1A***). The normalised fluorescence (blue curve) was obtained by dividing with the OD$_{600}$ (red curve) at each time point. (**C**) Values of fluorescence intensity at OD$_{600}$ = 0.05, Δt and Δf calculated using the kinetic profiles in (**B**). The parameters Δt and Δf are defined by the equations reported on top of the bar graphs and are graphically represented in (**B**). Averages of three independent replicates are shown with error bars representing standard errors.

medium (***Figure 2B***, left panel) a transient increase of normalised fluorescence of the culture reached a maximum level, followed by a subtle decline until the end of the experiment. The maximum fluorescence occurred approximately 2 hr after the culture had entered a starvation-induced growth arrest. Importantly, upon substitution of the *dnaA* 5'UTR-$N_t$ with the artificial non-nutritionally controlled 5'UTR$_{6/13}$ (***Osterman et al., 2013***), eGFP continued to accumulate throughout the starvation phase, while showing similar normalised culture fluorescence during exponential growth as the 5'UTR$_{6/13}$ strain (OD$_{600}$ = 0.05; ***Figure 2C***, left panel). This result demonstrates that the observed block of eGFP accumulation from 2 hr after entering growth arrest strongly depends on the presence of the *dnaA* 5'UTR-$N_t$.

To quantitatively describe the different kinetics of eGFP accumulation, we defined two parameters, $\Delta t$ and $\Delta f$. The first expresses the time difference (in hours) between the maxima of the growth and the fluorescence curves (*Figure 2B*), providing a measure of the time required to arrest eGFP accumulation in response to carbon exhaustion. The 5'UTR-$N_t$ strain gave a much smaller $\Delta t$ value than the 5'UTR$_{6/13}$ strain (*Figure 2C*, middle panel), suggesting efficient downregulation of eGFP expression in response to carbon starvation only when the reporter gene is under the control of the *dnaA* leader. The second parameter, $\Delta f$, denotes the relative accumulation of eGFP in the time interval between the culture reaching its maximum $OD_{600}$ and the termination of the measurement (24 hr). In this time frame, the 5'UTR$_{6/13}$ strain exhibited a clear increase in fluorescence, resulting in a significantly higher $\Delta f$ value compared to the 5'UTR-$N_t$ strain, which is characterised by a slightly negative $\Delta f$ value (*Figure 2C*, right panel).

## Stem P5 has an inhibitory effect on DnaA translation, but does not mediate DnaA downregulation at the onset of carbon starvation

Having established an in vivo reporter system that monitors post-transcriptional regulation of *dnaA* at the onset of carbon starvation, we performed a mutational analysis to identify sequence and structural elements required for the starvation-induced downregulation of DnaA. We first focused on stem P5 and its effect on translation rates in response to nutrient availability. A set of six mutant strains was constructed, with point mutation in the 5'UTR predicted to modify the stability of stem P5 (*Figure 3A*). As expected, the stabilising mutation G125C ($\Delta G_{folding}^{P5} = -11.3$ kcal/mol) completely abolished eGFP expression, whereas mutations that weaken stem P5 increased eGFP expression during exponential growth (*Figure 3B*). In particular, the C127A mutation, predicted to both break one base-pair and entail fraying at the bottom of the stem, caused a strong increase in basal eGFP expression. Most strikingly, none of the five stem P5 mutations affected the pattern of eGFP accumulation during the carbon starvation phase, as reflected by largely unchanged $\Delta t$ and $\Delta f$ values compared to the strain with the 5'UTR-$N_t$ wild-type sequence (*Figure 3C and D* and *Figure 3—figure supplement 1A*). Therefore, even though stem P5 stability affects in vivo translation efficiency per se, it is not involved in mediating the nutritional downregulation of DnaA. Additionally, these data suggest that the AGG sequence (nt 143–145), located 10 nt upstream of the AUG start codon, serves as a short and weak SD-like sequence.

To search for other regulatory sequences or structural determinants in the *dnaA* 5'UTR, we engineered a series of truncation mutants in which, starting from the 5'-end of the *dnaA* transcript, increasing portions of the 5'UTR were removed (*Figure 3E*). Although truncations T1–T4 increased eGFP expression levels, none of them notably changed $\Delta t$ or $\Delta f$ values (*Figure 3F–H* and *Figure 3—figure supplement 1B*). This result demonstrates that most of the leader is dispensable for the nutritional downregulation of DnaA and strengthens the conclusion that the putative CUG start codon (see toeprint in *Figure 1D*) does not impact DnaA translation in vivo. Adding more support, mutation of the canonical AUG start codon or deletion of the entire stem P5 completely abolished DnaA translation, even when the upstream in-frame CUG codon was converted to AUG (*Figure 3—figure supplement 2*). Hence, the putative upstream translation start site is not functional in vivo.

Two additional deletions (T5, T6) were created to assess whether the loop region of P5 plays a role in regulation of DnaA translation (*Figure 3—figure supplement 3*). Truncation T5 showed an eGFP accumulation pattern comparable to truncations T1–T4 and the wild-type construct. The T6 mutation removes nearly the entire 5'UTR and corresponds to the $\Delta UTR_{dnaA}$ mutation which, in our earlier study, suggested the *dnaA* 5'UTR to be required for starvation-induced downregulation of DnaA (*Leslie et al., 2015*). This mutant gave higher $\Delta t$ and $\Delta f$ values compared to the 5'UTR-$N_t$ control. However, it also exhibited much lower eGFP levels, indicative of inefficient translation initiation, presumably due to the absence of an extended single-stranded RNA stretch upstream of the AGG SD-like sequence (*de Smit and van Duin, 2003*). Even drastic changes in P5 loop nucleotides (mutants L1–L3) failed to cause substantial changes in $\Delta t$ and $\Delta f$ compared to 5'UTR-$N_t$ (*Figure 3—figure supplement 3*). Thus, the loop region of P5 does not affect regulation of DnaA translation.

In summary, stem P5 limits translation efficiency but does not mediate the starvation-induced downregulation of DnaA. Furthermore, most of the *dnaA* 5'UTR sequence is not required for the response to carbon starvation.

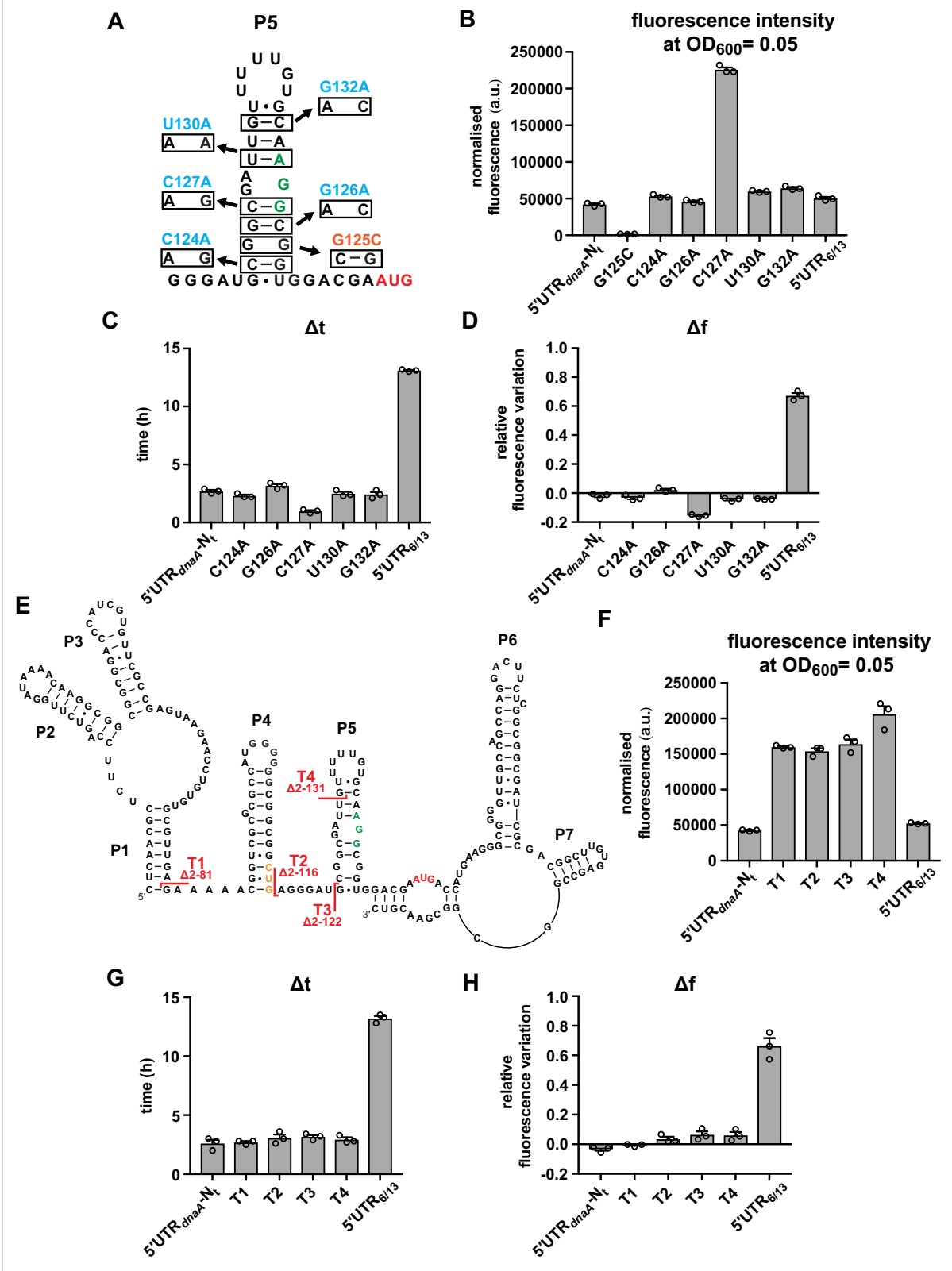

**Figure 3.** Most of the *dnaA* 5'UTR sequence is not required for the response to carbon starvation. (**A**) Stabilising (orange) and destabilising (light blue) mutations introduced in stem P5 by site-directed mutagenesis of the 5'UTR-$N_t$ reporter construct. The start codon and the putative RBS are indicated in red and green, respectively. (**B–D**) Values of fluorescence intensity at $OD_{600} = 0.05$, $\Delta t$ and $\Delta f$ calculated for the mutants in (**A**). (**E**) 5'UTR truncation mutants. Sites of truncation (**T1–T4**) are depicted on the mRNA secondary structure. The start codon and the putative Shine–Dalgarno (SD)-like

*Figure 3 continued on next page*

*Figure 3 continued*

sequence are indicated in red and green, respectively. The non-canonical CUG codon is coloured in orange. (**F–H**) Values of fluorescence intensity at $OD_{600}$ = 0.05, Δt and Δf calculated for the truncation mutants in (**E**). See *Figure 3—figure supplement 1A and B* for growth curves and fluorescence kinetic profiles. All data are shown as averages of three independent replicates with error bars representing standard errors.

The online version of this article includes the following figure supplement(s) for figure 3:

**Figure supplement 1.** Fluorescence kinetic profiles and growth curves of the reporter strains carrying mutations in the 5'UTR of the *dnaA* mRNA.

**Figure supplement 2.** The reported AUG start codon is the only functional translation start site in vivo.

**Figure supplement 3.** The loop region of P5 does not have a role in DnaA translation regulation.

## A sequence element downstream of the *dnaA* start codon mediates nutritional control of translation

Since most putative regulatory elements in the 5'UTR were ruled out as causative for starvation-induced regulation, we considered the mRNA region downstream of the translation start site, which codes for the N-terminal 26 amino acid residues of DnaA ($N_t$ hereafter). Strikingly, upon deletion of the $N_t$ region (*Figure 4A*, 5'UTR-$\Delta N_t$), eGFP fluorescence continued to accumulate during the carbon starvation phase (*Figure 4B*, upper panel). This resulted in significantly increased Δt and Δf values, indicating that this mRNA segment modulates *dnaA* translation at the onset of starvation. The repressing effect of $N_t$ was independent of the upstream 5'UTR (*Figure 4B and C*); strains with unrelated 5'UTRs (5'UTR$_{6/13}$ or 5'UTR$_{lac}$) showed similar $N_t$-dependent changes in the fluorescence kinetic profiles. While deletion of the $N_t$ region abolished the nutritional control of DnaA translation, duplication of this region in tandem (2 × $N_t$) resulted in an earlier response to starvation, as reflected in lower Δt and Δf values compared to the 5'UTR-$N_t$ control strain (*Figure 4D and E* and *Figure 4— figure supplement 1*). These results demonstrate that the mRNA region downstream of the AUG start codon is required and sufficient for arresting eGFP accumulation during the carbon starvation phase.

Because the $N_t$ region is translationally fused to eGFP, the stability of the reporter protein might be affected. Therefore, we compared eGFP protein stability between the 5'UTR-$N_t$ and 5'UTR-$\Delta N_t$ strains at the onset of starvation, after chloramphenicol-induced translation shutdown followed by western blotting and plate reader-based measurements of fluorescence. Both assays showed that the eGFP reporter was stable in both strains (*Figure 4—figure supplement 2*), excluding that the presence of DnaA $N_t$ at the N-terminus of eGFP significantly affects protein stability. This result is also congruent with a previous study which showed that the N-terminus of DnaA is required but not sufficient for DnaA proteolysis (*Liu et al., 2019*). Additionally, the plate reader experiment showed that the increase in fluorescence in the 5'UTR-$\Delta N_t$ strain during the carbon starvation phase was completely abolished in the presence of chloramphenicol (*Figure 4—figure supplement 2B and C*), confirming that the fluorescence increase in this strain depends on ongoing translation during starvation. Consistent with our previous results (*Figure 4B and C*), this effect was independent of the choice of 5'UTR.

Interestingly, a regulatory role of $N_t$ on DnaA translation in response to nutrient availability is also reflected in previously published ribosome profiling data, in which genome-wide translation profiles were analysed in the nutrient-rich medium PYE and the minimal medium M2G (*Aretakis et al., 2019*; *Schrader et al., 2014*). Our previous work showed that in these two media the nutrient-dependent regulation of DnaA translation is recapitulated, resulting in 4–5 times higher DnaA levels in PYE compared to M2G in exponentially growing cultures (*Leslie et al., 2015*). Consistently, according to calculations by Aretakis et al., the absolute number of translated DnaA molecules as well as the percentage of translation dedicated to DnaA synthesis is higher in PYE than in M2G (*Figure 4—figure supplement 3A and B*), demonstrating that DnaA is more efficiently translated during growth in rich medium. This is in contrast to the translational output of other cell cycle master regulators that is largely unaffected by the growth medium (*Figure 4—figure supplement 3A and B*). Most interestingly, when visualising the ribosome footprint density along the *dnaA* mRNA, we observed, in M2G, a strong enrichment of ribosomes footprints in the region encoding $N_t$ but only very low occupancy in the remaining part of the mRNA. By contrast, in PYE, ribosome footprints were more evenly distributed along the length of the *dnaA* mRNA (*Figure 4—figure supplement 3C and D*). Together this suggests that ribosomes pause in the region encoding the N-terminal amino acids of DnaA in a growth medium-dependent manner, thus supporting a direct involvement of the $N_t$ mRNA region in the regulation of DnaA translation in response to reduced nutrient availability.

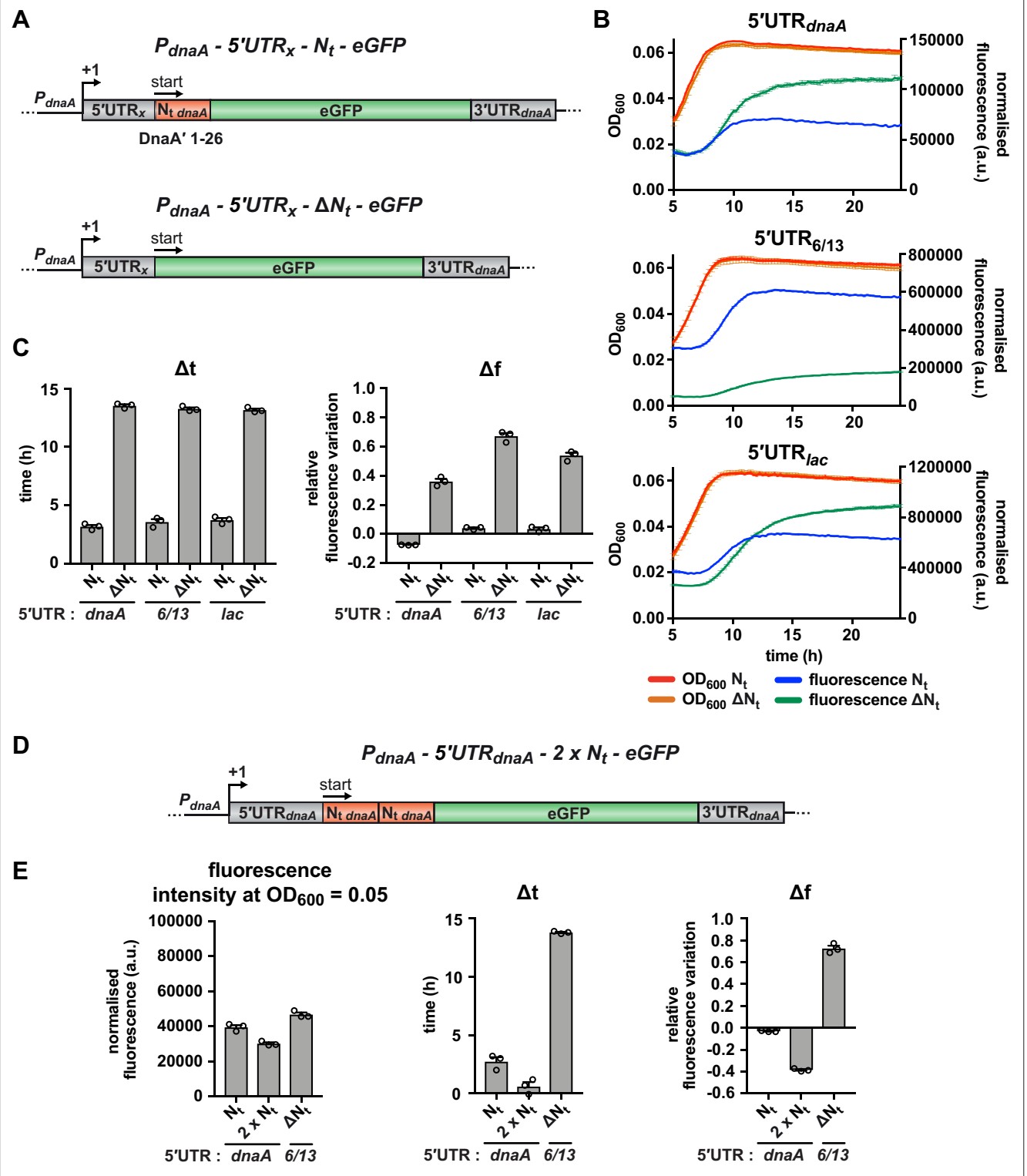

**Figure 4.** The mRNA region encoding the N-terminus of DnaA mediates the nutritional control of translation independently of the choice of the 5'UTR. (**A**) Schematic illustration of the reporter constructs utilised to study the role of the mRNA region encoding DnaA $N_t$ in the translational response to carbon starvation. In the 5'UTR$_x$-$N_t$ constructs (top), the *dnaA* promoter ($P_{dnaA}$) was fused to one of three possible 5'UTRs (5'UTR$_x$ = 5'UTR$_{dnaA}$, 5'UTR$_{6/13}$, and 5'UTR$_{lac}$; grey) followed by the first 26 codons of *dnaA* open reading frame ($N_{t\,dnaA}$; orange) and the eGFP gene (green). In the 5'UTR$_x$-$\Delta N_t$ constructs (bottom), the $N_t$ module is absent. (**B**) Growth curves and fluorescence kinetic profiles of the 5'UTR$_x$-$N_t$ (red and blue) and 5'UTR$_x$-$\Delta N_t$ (orange and green) constructs. (**C**) $\Delta t$ and $\Delta f$ values of the 5'UTR$_x$-$N_t$ and 5'UTR$_x$-$\Delta N_t$ reporter constructs. (**D**) The 5'UTR$_{dnaA}$-2 x $N_t$ reporter construct was obtained by

*Figure 4 continued on next page*

*Figure 4 continued*

duplicating the $N_t$ module (orange). (**E**) Values of fluorescence intensity at $OD_{600} = 0.05$, $\Delta t$ and $\Delta f$ calculated for the 2 x $N_t$ strain. The relative growth curves and fluorescence kinetic profiles are shown in *Figure 4—figure supplement 1*. All data are shown as averages of three independent replicates with error bars representing standard errors.

The online version of this article includes the following figure supplement(s) for figure 4:

**Figure supplement 1.** Fluorescence kinetic profile and growth curve of the 2 x $N_t$ reporter strain.

**Figure supplement 2.** The presence of DnaA $N_t$ at the N-terminus of eGFP does not change protein stability.

**Figure supplement 2—source data 1.** Source data for *Figure 4—figure supplement 2*.

**Figure supplement 3.** Ribosome profiling data indicating a role of $N_t$ in the growth medium-dependent regulation of DnaA translation.

## Downregulation of DnaA translation occurs during the early elongation phase and requires $N_t$'s native reading frame

We considered two alternatives by which $N_t$ might influence DnaA translation. Either the secondary structure of the mRNA segment encoding $N_t$ (i.e., P6 and P7) inhibits translation (*Bentele et al., 2013*; *Kudla et al., 2009*) or the nature of the codons or amino acids encoded by this sequence affects translation elongation. To discriminate between these possibilities, we constructed an $N_t$ double frameshift mutant (dfs$N_t$) by deleting one guanosine located at the beginning of the sequence encoding $N_t$ (position G164) and at the same time inserting a guanosine at the end of this sequence (after residue C233; *Supplementary file 1*). These two point mutations entirely change $N_t$ codon and peptide sequence between amino acid residues 3 and 26 while preserving the mRNA secondary structure and the eGFP reading frame (*Figure 5A*).

Strikingly, similar to the 5'UTR-$\Delta N_t$ strain, this mutant continued to synthesise eGFP also during the carbon starvation phase, reflected in high $\Delta t$ and $\Delta f$ values compared to the 5'UTR-$N_t$ strain (*Figure 5B and C*). This result argues against a role of secondary structure elements present in $N_t$ mRNA and instead suggests that the maintenance of the native reading frame is essential for nutritional control of translation.

We next tested whether the inhibitory effect of $N_t$ was maintained when this region was moved further downstream within the *dnaA* coding region or depended on its close proximity to the translation start site. For this, we made use of the tandem 2 x $N_t$ strain and disrupted the first $N_t$ region by introducing the two frameshift mutations to generate a dfs$N_t$-$N_t$ sequence (*Figure 5A*). Despite a reduction in basal eGFP expression, this strain showed clearly reduced $\Delta t$ and $\Delta f$ values (*Figure 5C* and *Figure 5—figure supplement 1*). This indicates starvation-dependent downregulation of *dnaA* and suggests that the $N_t$ sequence inhibits translation elongation even when it is located further downstream within the open reading frame.

In conclusion, our data are compatible with a model in which DnaA translation is downregulated under carbon starvation during the early stages of polypeptide chain elongation due to the specific amino acid or codon composition of $N_t$.

## A specific nascent chain amino acid sequence inhibits translation elongation under starvation

Having established that the $N_t$-mediated downregulation of DnaA translation depends on the native reading frame, we next wanted to identify sequence features in $N_t$ that are critical for regulation. A BLAST search revealed high conservation of the DnaA $N_t$ amino acid sequence in several members of the *Caulobacteraceae* family (*Figure 5—figure supplement 2A*). A general feature of the identified sequences is a high degree of hydrophobicity, with alanine being the most abundant amino acid. Using our fluorescent reporter, we designed two mutants in which regions of DnaA $N_t$ were deleted; mutant D1 lacks a hydrophobic region spanning residues 6–15, and mutant D2 lacks residues 18–22 (*Figure 5D*). Both D1 and D2 resulted in moderately higher $\Delta t$ and $\Delta f$ values (*Figure 5F* and *Figure 5—figure supplement 2B*) compared to the 5'UTR-$N_t$ strain, indicating a delay in starvation-dependent inhibition of DnaA translation. However, none of these mutations recapitulated the phenotype observed in the $\Delta N_t$ or dfs$N_t$ strains. In order to identify shorter motifs responsible for the phenotype observed in the D1 mutant, we generated additional amino acid deletions and substitutions in this segment of DnaA $N_t$. Most of these did not affect eGFP accumulation during carbon

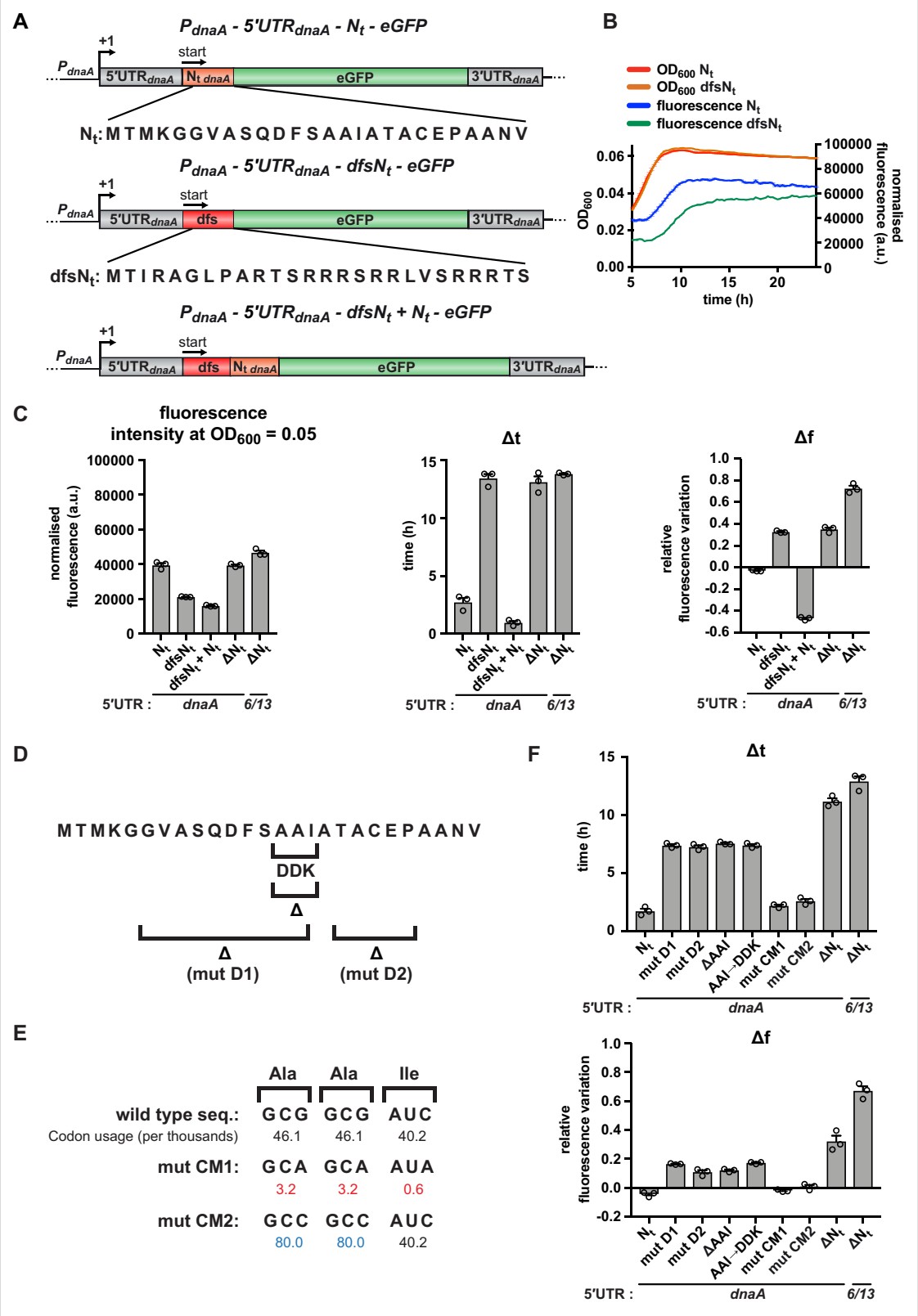

**Figure 5.** A specific amino acid sequence motif in the N-terminus of DnaA inhibits translation elongation under carbon starvation. (**A**) Schematic illustration of the double frameshift mutant reporter constructs. Top: wild-type amino acid sequence of DnaA $N_t$ in the context of the 5'UTR$_{dnaA}$-$N_t$ reporter construct. Middle: the double frameshift mutation alters the reading frame of $N_t$ (dfsN$_t$ module, in red), without affecting the eGFP frame. Bottom: the 5'UTR$_{dnaA}$-dfsN$_t$+ $N_t$ construct was generated by introducing the aforementioned double frameshift mutation in the first $N_t$ module of the 2 x

*Figure 5 continued on next page*

*Figure 5 continued*

$N_t$ construct (**Figure 4D**). (**B**) Comparison of fluorescence kinetic profiles between the 5'UTR-$N_t$ (blue) and the dfs$N_t$ (green) reporter strains. (**C**) Values of fluorescence intensity at $OD_{600}$ = 0.05, Δt and Δf calculated for the reporter strains in (**A**). (**D**) Illustration of deletions D1, D2, and ΔAAI, and the amino acid substitution AAI→DDK. (**E**) Illustration of the AAI motif codon mutations CM1 and CM2. Top: the wild-type codon sequence encoding the AAI motif, along with the values of codon usage. In mutant CM1 and CM2, the AAI motif is encoded by codons that present either lower (values in red) or higher (values in blue) usage. (**F**) Values of Δt and Δf calculated for the mutant reporter strains illustrated in (**D**) and (**E**). Data are reported as averages of three independent replicates with error bars representing standard errors.

The online version of this article includes the following figure supplement(s) for figure 5:

**Figure supplement 1.** Fluorescence kinetic profile and growth curve of the dfs$N_t$+ $N_t$ reporter strain.

**Figure supplement 2.** Fluorescence kinetic profiles and growth curves of the reporter strains carrying mutations in DnaA $N_t$.

**Figure supplement 3.** Additional mutations in the $N_t$ amino acid sequence.

starvation (**Figure 5—figure supplement 3**). However, the deletion or substitution of one group of three residues (A14-A15-I16; AAI motif hereafter) in the conserved hydrophobic region significantly increased both Δt and Δf values compared to the 5'UTR-$N_t$ control, similar to the 5'UTR-Δ$N_t$ and 5'UTR-dfs$N_t$ strains (**Figure 5F** and **Figure 5—figure supplement 2B**). These results indicate that the amino acids of the AAI motif contribute to the inhibition of DnaA translation.

Previous work established that synonymous codons, decoded by distinct isoacceptor aminoacyl (aa)-tRNAs (i.e., charged with the same amino acid), can display differences in their sensitivity to starvation (**Elf et al., 2003**; **Subramaniam et al., 2014**). For instance, the charged levels of major aa-tRNAs, which read the most abundant codons, rapidly decrease upon amino acid starvation (**Elf et al., 2003**). However, when the AAI motif codons were mutated to either less or more abundant synonymous codons (**Figure 5E**), no significant effects on the starvation-dependent inhibition of DnaA translation were observed (**Figure 5F** and **Figure 5—figure supplement 2B**). This suggests that the amino acid sequence of the N-terminus of DnaA mediates the starvation-induced translation arrest irrespective of the synonymous codon choice. It is of note that the introduction of less abundant codons, containing A or T in the third position, resulted in higher eGFP expression levels (**Figure 5—figure supplement 2B**), probably due to a decreased propensity for a stable mRNA structure near the translation start site (**Bentele et al., 2013**; **Goodman et al., 2013**; **Kudla et al., 2009**).

Thus, our data suggest that, under carbon-limiting conditions, translation elongation efficiency decreases in a sequence-dependent manner during the synthesis of the DnaA N-terminus. Importantly, this effect depends on the amino acid composition of the DnaA $N_t$, but not the codons encoding it.

## The sequence-specific translation downregulation mechanism operates in a heterologous host

We wondered whether the identified starvation-responsive sequence element also can function in a different cellular context and chose to analyse the effect of the $N_t$ region of the *Caulobacter dnaA* mRNA in *E. coli*. For this, we transcriptionally fused the 5'UTR-$N_t$-eGFP or 5'UTR-Δ$N_t$-eGFP modules, respectively, from the *Caulobacter* reporter to an IPTG-inducible Lambda O1 promoter on a low-copy plasmid and transformed the resulting construct into *E. coli* (**Figure 6A**). Indeed, shifting to carbon-limiting conditions, eGFP accumulation ceased completely in the strain with the *Caulobacter* $N_t$ sequence, while it continuously increased fluorescence in the strain lacking this region, resulting in higher Δt and Δf values (**Figure 6A and B**). Like in *Caulobacter*, the presence of $N_t$ did not cause destabilisation of the eGFP reporter in *E. coli* (**Figure 6—figure supplement 1A**). A repressing effect was also observed when the 5'UTR of *Caulobacter dnaA* was replaced by the artificial 5'UTR$_{6/13}$, demonstrating that $N_t$ is sufficient for regulating translation in response to carbon starvation also in *E. coli* (**Figure 6B** and **Figure 6—figure supplement 1B**). Finally, we introduced some of the relevant $N_t$ mutations in the *E. coli* reporter plasmid and verified that, similarly to what we observed in *Caulobacter*, they led to increased Δt and Δf values (**Figure 6B** and **Figure 6—figure supplement 1B**), strongly suggesting that $N_t$ acts through a conserved mechanism on translation elongation.

Together, these data show that the identified starvation-responsive sequence encoded within the N-terminus of DnaA constitutes an autonomous regulatory element that can operate in a heterologous host.

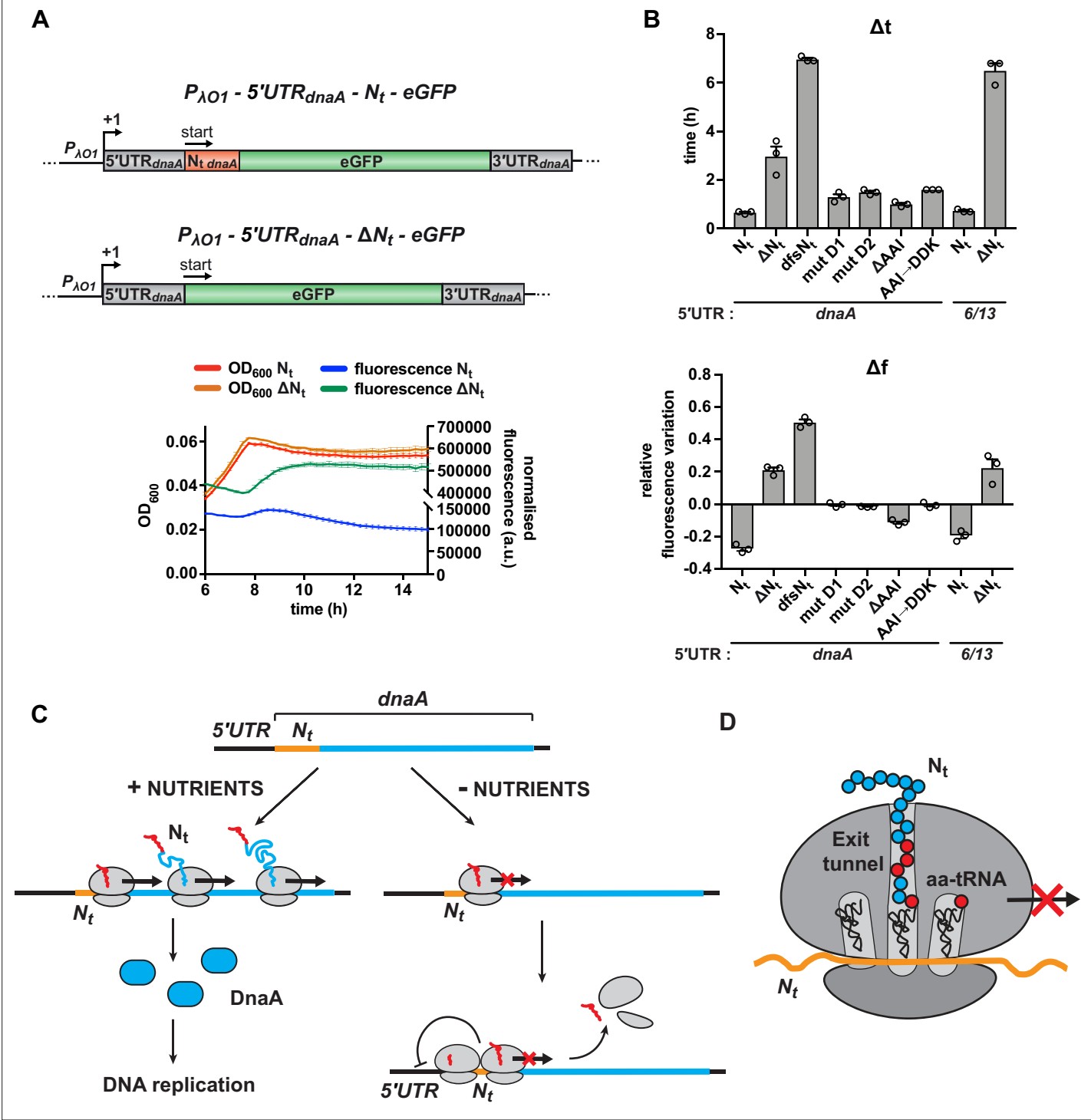

**Figure 6.** Starvation-induced inhibition of DnaA translation elongation by an autonomously operating nascent chain sequence element. (**A**) Schematic illustration of the reporter constructs utilised to study DnaA $N_t$ functionality in the heterologous host *E. coli*. In the 5'UTR$_{dnaA}$-$N_t$ constructs (top), the IPTG-inducible Lambda promoter O1 ($P_{\lambda O1}$) was fused to the 5'UTR$_{dnaA}$ (grey) followed by *Caulobacter*'s $N_t$ ($N_{t\,dnaA}$ - orange) and *eGFP* (green). The 5'UTR$_{dnaA}$-$\Delta N_t$ constructs (below) lack the $N_t$ module. The constructs were cloned into the low-copy plasmid pZE12 and transformed into *E. coli* MG1655. The *E. coli* reporter strains were grown for 15 hr in M9 medium supplemented with 0.02% glucose in a microplate reader. Normalised culture fluorescence was measured to monitor eGFP synthesis in the reporter strains (5'UTR$_{dnaA}$-$N_t$ versus 5'UTR$_{dnaA}$-$\Delta N_t$, lower panel). (**B**) Values of Δt and Δf calculated for the strains in (**A**), the $N_t$ mutants and the 5'UTR$_{6/13}$ reporter strains. The relative growth curves and fluorescence kinetic profiles are shown in *Figure 6—figure supplement 1*. Background subtraction, fluorescence normalisation, and Δt and Δf quantifications were performed as described for the *Caulobacter* reporter strains. Data are reported as averages of three independent replicates with error bars representing standard errors. (**C**)

*Figure 6 continued on next page*

*Figure 6 continued*

The proposed model for the regulation *dnaA* translation under carbon starvation. In the presence of nutrients (left-hand side), *dnaA* mRNA is efficiently translated, leading to accumulation of active DnaA (light blue) that, in turn, triggers DNA replication initiation. Under carbon starvation (right-hand side), translation rate decreases during the first stages of the elongation phase at a specific sequence located downstream of the translation start site ($N_t$; orange). The resulting ribosome traffic jams possibly stimulate disassembly of the leading ribosome from the mRNA and/or inhibit new cycles of translation initiation. The regulatory nascent polypeptide ($N_t$) is shown in red. (**D**) The downregulation of *dnaA* translation requires specific amino acid residues (in red) encoded in the region downstream of the start codon ($N_t$; orange). These specific amino acids arrest translation after or while being added to the growing DnaA nascent polypeptide ($N_t$).

The online version of this article includes the following figure supplement(s) for figure 6:

**Figure supplement 1.** The presence of DnaA $N_t$ at the N-terminus of eGFP does not change protein stability in the *E. coli* reporter.

## Discussion

The integration of nutritional information with essential cellular processes such as protein synthesis and DNA replication is critical for cellular life, but poorly understood on a molecular level. This study reveals a new mechanism, by which some bacteria can regulate the synthesis of the replication initiator DnaA in response to nutrient availability by modulating the rate of translation elongation. In the model organism *C. crescentus*, we identified a specific N-terminal sequence in the nascent DnaA polypeptide chain that acts as a novel regulatory element which tunes DnaA translation elongation in response to the cellular nutrient and growth status. With its crucial function in bacterial replication initiation, the abundance and activity of DnaA must be precisely regulated in response to environmental and cellular changes to ensure cellular survival.

### A model for DnaA regulation under carbon starvation

Based on our results, we propose a model of nutritionally controlled DnaA translation. Under optimal conditions, efficient transcription of *dnaA* and a high rate of DnaA translation ensure the accumulation of active DnaA that triggers DNA replication initiation (*Figure 6C*). At the onset of carbon starvation, translation ceases during the early stage of the elongation phase at a specific sequence located downstream of the *dnaA* start codon (as discussed below). Ribosome pausing at this sequence, as supported by our analysis of ribosome profiling data (*Figure 4—figure supplement 3*), is expected to result in ribosome traffic jams that, as previously proposed, stimulate premature translation termination either directly, by inducing disassembly of the leading ribosome from the mRNA, or indirectly, by recruiting the tmRNA or ArfB ribosome rescue systems that are present in *Caulobacter* (*Feaga et al., 2014*; *Ferrin and Subramaniam, 2017*; *Keiler et al., 2000*; *Subramaniam et al., 2014*). Due to the proximity of the stalling site and the start codon, persistent queues of 2–3 ribosomes might also decrease translation rates by physically preventing access of initiating ribosomes to the translation start site (*MacDonald et al., 1968*; *Mitarai et al., 2008*; *Tuller et al., 2010*). Additionally, ribosome pausing may result in increased mRNA decay or Rho-dependent termination of transcription (*Samatova et al., 2020*). The postulated starvation-dependent elongation pause significantly reduces the rate of DnaA synthesis. The concomitant degradation of full-length DnaA by Lon then aids the rapid clearance of DnaA from the cell (*Leslie et al., 2015*), preventing DNA replication initiation.

### The molecular mechanism of starvation-induced translation inhibition

The nutritional downregulation of DnaA translation depends on a specific sequence downstream of the *dnaA* start codon. Since neither the RNA secondary structure of this sequence nor the codon choice impacted the nutritional regulation of DnaA (*Figure 5*), we rule out that the RNA structure spatially hinders ribosome movement. Furthermore, we have shown that synonymous mutations of the AAI motif did not perturb the capacity of $N_t$ to mediate the regulatory response. Based on this result and the previously made findings that under starvation the charged levels of the different isoacceptor tRNAs depend on their codon usage during translation (*Dittmar et al., 2005*; *Elf et al., 2003*; *Subramaniam et al., 2014*), we consider it also unlikely that reduced charging levels of a specific set of isoacceptor tRNAs (e.g., Ala-tRNAs) is the major cause of the downregulation of translation during starvation. Instead, we suggest that the amino acids encoded by this sequence arrest translation after, or while, being added to the growing DnaA polypeptide (*Figure 6D*). This model is in line with the recent observation that the identity of the amino acids encoded by early codons of a gene can impact protein synthesis yield (*Verma et al., 2019*). Previous work has also demonstrated that certain nascent peptide sequences specifically arrest translation through interactions

with the ribosome exit tunnel surface, thereby regulating the expression of downstream genes through translational coupling, or affecting transcription termination and mRNA decay (*Ito and Chiba, 2013*; *Wilson et al., 2016*). A well-characterised example is the force-sensing arrest peptide SecM, which contributes to the translational control of the downstream gene *secA* that encodes a translocase subunit (*Murakami et al., 2004*; *Nakatogawa and Ito, 2002*). Another class of arrest peptides modulates gene expression by interacting with small effector molecules in the ribosome exit tunnel (*Gong et al., 2001*; *Herrero Del Valle et al., 2020*). The AAI motif identified in this study, possibly along with neighbouring amino acids in the N-terminus of DnaA, might interact with the ribosome in a similar way as arrest peptides. However, while arrest peptides commonly serve as *cis*-acting elements that regulate the expression of downstream genes, the sequence element that we describe represents, to our knowledge, the first example of a regulatory nascent polypeptide sequence in the N-terminus of a protein that directly modulates translation efficiency of the same protein in response to nutrient availability.

## Possible ways of sensing nutrients

The identified nascent sequence element specifically decreases overall DnaA translation rates at the onset of carbon starvation, demonstrating that it is responsive to the nutrient status of the cell. Since this is reca-pitulated in a heterologous host (*Figure 6B*), it argues for a sensing mechanism that is triggered by a global starvation-induced change in translation strategy (*Li et al., 2018*), rather than by *Caulobacter*-specific acces-sory factors. For example, reduction in the supply of one or more components of the translation machinery (e.g., ribosomes, aa-tRNAs, elongation factors, GTP) could in principle slow down overall translation rates (*Li et al., 2018*; *Samatova et al., 2020*; *Zhang et al., 2010*). Consequently, translating ribosomes could become more susceptible to starvation-specific inhibitory sequences akin to the sequence identified here. Alterna-tively, a highly conserved *trans*-acting factor, a metabolite, or a signalling molecule could be involved in the nutrient-sensing mechanism. Such a putative factor or small molecule might either release the translational arrest during nutrient-rich conditions or enhance it during starvation.

## The *dnaA* 5'UTR as a regulatory element

This work presents the first detailed structural and functional characterisation of the *dnaA* 5'UTR. Although the 5'UTR is not required for modulating translation rates under starvation, it might play other regulatory roles. Consistent with a previous study (*Cheng and Keiler, 2009*), our results show the inhibitory effect of the *dnaA* leader on translation per se and demonstrate that the efficiency of translation initiation can be signifi-cantly increased by reducing the stability of the secondary structure around the translation start site. Interest-ingly, although the *dnaA* 5'UTR, like the majority of mRNAs in *Caulobacter*, lacks an obvious SD sequence (*Schrader et al., 2014*), our results suggest that a short AGG motif buried in stem P5 could function as an SD-like sequence. The modulation of SD-like sequence accessibility by a *trans*-acting factor (e.g., a regula-tory RNA or protein) interfering with stem P5 structure could conceivably be a strategy to directly regulate the initiation of *dnaA* translation in response to specific external or internal cues. Consistent with this idea, a previous computational study proposed the *dnaA* 5'UTR of *Caulobacter* to be a possible regulatory target of multiple small non-coding RNAs (*Beroual et al., 2018*).

## Conclusions

In conclusion, we have identified a new sequence-dependent mechanism that modulates translation elon-gation of a specific protein in response to nutrient availability, highlighting the importance of the first stages of translation elongation as a regulatory target. It is particularly striking that the mechanism described func-tions to control the concentration of the highly conserved replication initiator DnaA. This protein is critical for DNA replication in nearly all bacteria, and nutritional control mechanisms that modulate DnaA synthesis contribute to the correct timing of DNA replication initiation in response to changes in nutrient availability and growth rate. The finding that DnaA in *Caulobacter* and related bacteria is regulated at the level of trans-lation elongation reveals important new layers of control acting on this important protein.

The conclusions reported here may suggest a more general pattern. We consider it possible that the proposed mechanism of translational downregulation via nascent peptide sequence-dependent effects potentially tunes the expression of other genes during starvation. Principally, this mechanism could help to globally remodel the proteome under starvation conditions when cells need to reallo-cate their resources from growth-promoting to maintenance functions.

# Materials and methods

## Key resources table

| Reagent type (species) or resource | Designation | Source or reference | Identifiers | Additional information |
|---|---|---|---|---|
| Gene (*Caulobacter crescentus*) | *dnaA*<br><br>CCNA_00008 | GenBank | YP_002515383.1 | |
| Strain, strain background (*Caulobacter crescentus*) | NA1000 | Michael Laub, Massachusetts Institute of Technology | | Electrocompetent cells |
| Strain, strain background (*Escherichia coli*) | DH5$\alpha$ | Michael Laub, Massachusetts Institute of Technology | | Chemical competent cells. |
| Strain, strain background (*Escherichia coli*) | MG1655 | Michael Laub, Massachusetts Institute of Technology | | |
| Antibody | Anti-DnaA (rabbit polyclonal) | *Jonas et al., 2011* | | (1:5,000) |
| Antibody | Anti-GFP (rabbit polyclonal) | Thermo Fisher Scientific | Cat# A11122<br><br>RRID: AB_221569 | (1:20,000) |
| Antibody | Anti-rabbit IgG (H + L) secondary antibody, HRP (goat polyclonal) | Thermo Fisher Scientific | Cat# 31460<br><br>RRID: AB_228341 | (1:5,000) |
| Recombinant DNA reagent | pMR10 (plasmid) | Michael Laub, Massachusetts Institute of Technology (*Roberts et al., 1996*) | | |
| Recombinant DNA reagent | pMCS (plasmid) | Martin Thanbichler, MPI Marburg (*Thanbichler et al., 2007*) | | |
| Recombinant DNA reagent | pZE12-luc (plasmid) | Victor Sourjik, MPI Marburg (*Lutz and Bujard, 1997*) | | |
| Sequence-based reagent | fw 1<br><br>(DNA oligonucleotide) | This paper<br><br>(synthesised by Sigma-Aldrich-Merck) | | Amplification of the DNA template for in vitro transcription.<br><br>GAAATTAATACGACTCACTATA GCTCAACGCTCTTCCAGTCTTGG |
| Sequence-based reagent | fw 2<br><br>(DNA oligonucleotide) | This paper<br><br>(synthesised by Sigma-Aldrich-Merck) | | Amplification of the DNA template for sequencing reaction.<br><br>CAAATATTTACAAAGGCCGATCAGGG |
| Sequence-based reagent | primer A<br><br>(DNA oligonucleotide) | This paper<br><br>(synthesised by Sigma-Aldrich-Merck) | | Probing, toeprinting, and amplification of the DNA template for in vitro transcription and sequencing reaction.<br><br>CACCCAGCTCACGCTTCAAAG |
| Sequence-based reagent | primer B<br><br>(DNA oligonucleotide) | This paper<br><br>(synthesised by Sigma-Aldrich-Merck) | | Probing.<br><br>GAGAAGTCCTGGCTGGCAAC |
| Chemical compound, drug | Kanamycin | Carl Roth | Cat# T832.5<br><br>CAS: 25389-94-0 | 1 µg/mL (PYE and M2G liquid)<br><br>30 µg/mL (LB liquid)<br><br>25 µg/mL (LB plates) |
| Chemical compound, drug | Ampicillin | Carl Roth | Cat# K029.4<br><br>CAS: 69-52-3 | 50 µg/mL (LB liquid)<br><br>100 µg/mL (LB plates) |
| Chemical compound, drug | Chloramphenicol | Carl Roth | Cat# 3886.1<br><br>CAS: 56-75-7 | 100 µg/mL (translation shut-off in *Caulobacter*)<br><br>200 µg/mL (translation shut-off in *E. coli*) |
| Chemical compound, drug | IPTG | Sigma-Aldrich | Cat# PHG0010<br><br>CAS 367-93-1 | 1 mM |
| Chemical compound, drug | [γ-P$^{32}$]-ATP (10 mCi/mL, 3000 Ci/mmol) | Perkin Elmer | Cat# BLU002A<br><br>CAS: 2964-07-0 | |

*Continued on next page*

*Continued*

| Reagent type (species) or resource | Designation | Source or reference | Identifiers | Additional information |
|---|---|---|---|---|
| | | | Cat# 215902 | |
| Chemical compound, drug | Pb²⁺ acetate | Sigma-Aldrich-Merck | CAS: 6080-56-4 | Chemical probe |
| Commercial assay or kit | SuperSignal Femto West reagent | Thermo Fisher Scientific | Cat# 34094 | |
| Commercial assay or kit | Calf Intestinal Alkaline Phosphatase (CIAP) | Invitrogen | Cat# 18009-019 | |
| Commercial assay or kit | T4 Polynucleotide Kinase (PNK) | Thermo Fisher Scientific | Cat# EK0031 | |
| Commercial assay or kit | RNase T1 | Invitrogen | Cat# AM2283 | |
| Commercial assay or kit | RNase V1 | Invitrogen | Cat# AM2275 | |
| Commercial assay or kit | Megascript Kit | Life Technologies | Cat# AM1330 | |
| Commercial assay or kit | SuperScript IV reverse transcriptase | Invitrogen | Cat# 18090010 | |
| Commercial assay or kit | Sequenase Cycle Sequencing Kit | Affymetrix | Cat# 78500 | |
| Software, algorithm | mFold | *Zuker, 2003* http://www.unafold.org/ | RRID: SCR_008543 | |
| Software, algorithm | ViennaRNA (RNAfold, RNAcofold) | *Gruber et al., 2008* http://rna.tbi.univie.ac.at/ | RRID:SCR_008550 | |
| Software, algorithm | CMfinder | Zasha Weinberg, Leipzig University (*Yao et al., 2006*) | | Version 0.4.1.4 |
| Software, algorithm | Fiji (ImageJ) | *Rueden et al., 2017* https://fiji.sc/ | RRID:SCR_002285 | |
| Software, algorithm | Image Lab | Bio-Rad https://www.bio-rad.com/en-ca/product/image-lab-software | RRID:SCR_014210 | Version 6.0 |
| Software, algorithm | NCBI BLAST (BLASTn, BLASTp, tBLASTn) | *Altschul et al., 1990* https://blast.ncbi.nlm.nih.gov/ | RRID:SCR_004870 | |
| Software, algorithm | mafft | *Katoh and Standley, 2013* https://mafft.cbrc.jp/alignment/software/ | RRID:SCR_011811 | Version 7.427 |
| Software, algorithm | IQTREE | *Nguyen et al., 2015* http://www.iqtree.org/release/v1.6.10 | | Version 1.6.10 |
| Software, algorithm | SeaView | *Gouy et al., 2010* http://doua.prabi.fr/software/seaview | RRID:SCR_015059 | Version 5.0.4 |
| Software, algorithm | R (including splines package) | R Project https://www.R-project.org/ | RRID:SCR_001905 | |
| Software, algorithm | GraphPad Prism | https://www.graphpad.com | RRID:SCR_002798 | Version 7.0 |
| Software, algorithm | Bowtie | *Langmead et al., 2009* http://bowtie-bio.sourceforge.net/index.shtml | RRID:SCR_005476 | Version 1.2.2 |
| Software, algorithm | SAMtools | *Li et al., 2009* http://htslib.org/ | RRID:SCR_002105 | |
| Software, algorithm | Integrative Genomics Viewer (IGV) | *Robinson et al., 2011* http://www.broadinstitute.org/igv/ | RRID:SCR_011793 | Version 2.10.2 |
| Other | Yeast tRNA | Invitrogen | Cat# AM7119 | |
| Other | *E. coli* 30S subunit | *Romilly et al., 2019* | | |

*Continued on next page*

*Continued*

| Reagent type (species) or resource | Designation | Source or reference | Identifiers | Additional information |
|---|---|---|---|---|
| Other | *E. coli* initiator tRNA$^{fMet}$ | *Romilly et al., 2019* | | |
| Other | 12% Mini-PROTEAN TGX Stain-Free gels | Bio-Rad | Cat# 4568046 | |
| Other | Trans-Blot Turbo System | Bio-Rad | Cat# 1704150EDU | |
| Other | LI-COR Odyssey Fc imaging system | LI-COR | | https://www.licor.com/bio/odyssey-fc/ |
| Other | G-50 Microspin columns | GE Healthcare | Cat# 27-5330-01 | |
| Other | Personal Molecular Imager (PMI) system | Bio-Rad | Cat# 170-9400 | |
| Other | 96-well transparent plates with flat-bottom and lid | Greiner Bio-One | Cat# 655182 | |
| Other | Spark multimode microplate reader | TECAN | | https://lifesciences.tecan.com/multimode-plate-reader |

## Growth conditions

All *Caulobacter* reporter strains were derived from the wild-type strain NA1000 and were grown routinely in PYE or M2G (minimal medium with 0.2% glucose) at 30 °C while shaking at 200 rpm. For the carbon exhaustion experiments, exponentially growing cells were washed and inoculated in M2G$_{1/10}$ (0.02% glucose). *E. coli* DH5α was used as a host for plasmid construction and was routinely grown in LB medium at 37 °C while shaking at 200 rpm. The *E. coli* reporter strains were derived from wild-type strain MG1655. Pre-cultures were grown at 30 °C while shaking at 200 rpm in LB or M9 minimal medium supplemented with 0.2% glucose. Carbon exhaustion experiments in the plate reader were performed in M9 medium supplemented with 0.02% glucose and 1 mM IPTG. When appropriate, the following antibiotic concentrations were used: 1 µg/mL kanamycin (PYE and M2G liquid media), 25 µg/mL kanamycin (PYE plates), 30 µg/mL kanamycin (LB liquid medium), 50 µg/mL kanamycin (LB plates), 50 µg/mL ampicillin (LB liquid medium), and 100 µg/mL ampicillin (LB plates).

## Bacterial strains and plasmid construction

All the plasmids in this study were constructed by Gibson assembly (*Gibson and Young, 2009*). pMR10-BG (*Supplementary file 1A*) was derived from pMR10 (*Roberts et al., 1996*) by exchanging *lacZα* and *oriT* with a multicloning site. The vector backbone for the pMR10-5'UTR-N$_t$-GFP (*Supplementary file 1B*) construct was obtained upon digestion of pMR10-BG with HindIII and BamHI, while the insert was first assembled in a smaller vector, pMCS (*Thanbichler et al., 2007*), and then amplified by PCR with appropriate primers. The insert comprised the following sequences, in order, (1) the *rrnB1*-T1T2 terminator, (2) 245 bp upstream of the *dnaA* transcription start site (P$_{dnaA}$), (3) the 5'UTR$_{dnaA}$, (4) the first 78 bp of the *dnaA* open reading frame, (5) the eGFP gene (fused in-frame), and (6) 63 bp downstream of the *dnaA* stop codon (3'UTR$_{dnaA}$). The other pMR10 reporter constructs (*Supplementary file 1E*) were generated by site-directed mutagenesis using pMR10-5'UTR-Nt-GFP as template. All constructs were finally transformed into *C. crescentus* NA1000 by electroporation.

Plasmid pZE12-BG (*Supplementary file 1C*) was generated from pZE12-luc plasmid (*Lutz and Bujard, 1997*) by deleting the luciferase gene and part of the P$_{λO1}$ promoter. The pZE12 reporter plasmids were obtained by replacing the luciferase gene in the pZE12-luc plasmid, with the appropriate 5'UTR-N$_t$-eGFP module amplified from the related pMR10 construct (*Supplementary file 1D*). The pZE12 reporter plasmids were transformed into *E. coli* MG1655 as previously described (*Chung et al., 1989*). All constructs were confirmed by DNA sequencing. See *Supplementary file 1E* for the sequence of the mutants and *Supplementary file 2* for the complete list of plasmids and strains.

## Immunoblotting

Pelleted cells were resuspended in 1× SDS sample buffer (1/10 volume of β-mercaptoethanol added before use), normalised to the optical density of the culture (40 µL per units of 0.1 OD$_{600}$), and heated to 95 °C for 10 min. Protein extracts were subjected to SDS-PAGE for 90 min at 130 V at room temperature on 12% Mini-PROTEAN TGX Stain-Free gels (Bio-Rad #4568046). Samples were transferred to nitrocellulose membranes using a Bio-Rad Trans-Blot Turbo system ('standard sd'

protocol). To verify equal loading, total protein was visualised using a Bio-Rad Gel Doc imager after PAGE separation and prior to blotting. DnaA and eGFP were detected with anti-DnaA (*Jonas et al., 2011*; 1:5,000 dilution) and anti-GFP (Thermo Fisher Scientific #A11122, 1:20,000 dilution) primary antibodies, respectively, followed by a 1:5,000 dilution of goat anti-rabbit secondary horseradish peroxidase-conjugated antibody (Thermo Fisher Scientific #31460). After treating the membranes with the SuperSignal Femto West reagent (Thermo Fisher Scientific #34094), blots were scanned with a LI-COR Odyssey Fc imaging system. Fiji (ImageJ) was used for image processing and band intensity quantification (*Rueden et al., 2017*).

## In vitro transcription of RNA

A DNA template containing a T7 promoter was generated by PCR using *C. crescentus* NA1000 genomic DNA as template and the following primers: 5'-GAAAT<u>TAATACGACTCACTATAG</u>CTCA ACGCTCTTCCAGTCTTGG-3' (fw1, forward — T7 promoter underlined) and 5'-CACCCAGCTCAC GCTTCAAAG-3' (primer A, reverse). The Megascript Kit (Life Technologies, #AM1330) was used for in vitro transcription. The RNA was purified on a 6% denaturing polyacrylamide gel. The band corresponding to the transcribed RNA was detected by UV-shadowing and cut from the gel. The RNA was then eluted into 300 mM Na-acetate, 0.1% SDS, and 1 mM EDTA. After phenol-chloroform-isoamyl alcohol (25:24:1) extraction and ethanol precipitation, the RNA pellet was dissolved in water. RNA concentration was measured by NanoDrop and the quality was assessed by denaturing PAGE. 5'-end-labeling of CIAP-treated RNA (Invitrogen, #18009-019) with T4 PNK (Thermo Fisher Scientific, #EK0031) in buffer A and [$\gamma$-P$^{32}$]-ATP (10 mCi/mL, 3000 Ci/mmol; Perkin Elmer # BLU002A). The labelled RNA was gel-purified as described above.

## Native gel electrophoresis

The in vitro transcribed RNA was refolded for 1 min at 95 °C, followed by 1 min on ice and 10 min at 30 °C in 1 × TMK buffer (100 mM K$^+$-acetate, 5 mM Mg$^{2+}$-acetate, 50 mM Tris-HCl pH 7.5). The sample (about 10,000 cpm) was directly loaded on a 6% non-denaturing polyacrylamide gel. The gel was run at 4 °C for 2 hr (200 V) in 0.5× TEB, then covered with plastic film, and exposed to a storage phosphor screen for 1 hr. Signals were detected using a PMI system (Bio-Rad) and analysed by Image Lab 6.1 (Bio-Rad).

## Structural probing

Two probing approaches were used to determine the secondary structure of the in vitro transcribed RNA. In the first, 0.1 pmol of $^{32}$P-5'-end-labelled RNA were subjected to RNase T1 or Pb$^{2+}$ probing at 30 °C in the presence of 2 µg carrier yeast tRNA (Invitrogen #AM7119) (*Figure 1—figure supplement 1B*). The labelled RNA was first denatured for 1 min at 95 °C, followed by 1 min on ice. Before adding the carrier tRNA and the probes, the labelled RNA was allowed to refold for 10 min in a buffer containing 10 mM Tris-acetate pH 7.5, 100 mM K$^+$-acetate, 5 mM, DTT, and 10 mM Mg$^{2+}$-acetate (10 µL total reaction volume). RNase T1 probing was done using one unit of enzyme (Invitrogen, #AM2283) for 5 min, and stopped by adding 40 µL cold 0.3 M Na$^+$-acetate. Chemical probing was done with Pb$^{2+}$-acetate (Sigma-Aldrich-Merck) at a final concentration of 10 mM and 25 mM for 5 min. Reactions were stopped with addition of cold EDTA (50 mM final concentration) followed by 35 µL of cold 0.3 M Na$^+$-acetate. After phenol-chloroform-isoamyl alcohol extraction, RNA was ethanol-precipitated. RNA pellets were dried and dissolved in loading dye. About 10,000 cpm of each sample were loaded on a 12% denaturing polyacrylamide gel, fixed for 5 min (10% ethanol, 6% acetic acid), and transferred to 3 mm Whatman paper. Signals were detected using a storage phosphor screen and a PMI system (Bio-Rad), and analysed using Image Lab 6.1 (Bio-Rad). Cleavage pattern induced by the probes was assigned using RNase T1 hydrolysis under denaturing conditions (i.e., T1 ladder) and an alkaline ladder. The ladders were prepared following the manufacturer's instructions (RNase T1 manual, Invitrogen). We used 0.2 pmol of 5'-end-labelled RNA/2 µg of carrier yeast tRNA for the T1 ladder, and 0.1 pmol of 5'-end-labelled RNA/1 µg of carrier yeast tRNA were used for the alkaline ladder.

The second probing approach used primer extension to detect cleavage positions after treatment with RNases T1 and V1 (*Figure 1—figure supplement 1C and D*). Here, reactions were performed as described above, but using 1 pmol of unlabelled RNA and 1 µg carrier yeast tRNA. RNase V1 probing

was done using 0.1, 0.02, or 0.01 enzyme units (Invitrogen, #AM2275) for 5 min, stopped by adding 40 µL of cold 0.3 M Na$^+$-acetate. T1 ladders were done in the presence of 1 pmol of unlabelled, denatured RNA and 2 µg of carrier yeast tRNA. The probed RNA and the ladders were phenol-chloroform-isoamyl alcohol extracted and precipitated. Dried pellets were then resuspended in water, and primer extension was performed at 37 °C for 20 min using 100 U of SuperScript IV reverse transcriptase (Invitrogen, #18090010) and two different $^{32}$P-5'-end-labelled primers: 5'-CACCCAGCTCACGCTT CAAAG-3' (primer A) or 5'-GAGAAGTCCTGGCTGGCAAC-3' (primer B). Primers were labelled using T4 PNK in buffer A and [γ-P$^{32}$]-ATP (10 mCi/mL, 3000 Ci/mmol) and purified using G-50 Microspin columns (GE Healthcare, #27-5330-01). Reverse transcription reactions were stopped by adding 40 µL of cold 0.3 M Na$^+$-acetate. After phenol-chloroform extraction, the RNA template was removed by addition of 3 M KOH for 3 min at 90 °C, followed by 1 hr incubation at 37 °C. After base neutralisation with HCl, the cDNAs were ethanol-precipitated, centrifuged, dissolved in loading buffer, and resolved on 12% polyacrylamide gels. Gels were fixed and treated as above. The sequencing reactions were done using the USB Thermo Sequenase Cycle Sequencing Kit (Affymetrix #78500) and $^{32}$P-5'-end-labelled primers A and B. The template for the sequencing reaction was obtained by PCR using *C. crescentus* NA1000 genomic DNA and the following primers: 5'-CAAATATTTACAAAGGCCGA TCAGGG-3' (fw2, forward) and primer A (reverse).

## Toeprinting assay

The toeprinting assay (*Huttenhofer and Noller, 1994*) was performed in a 10 µL reaction volume using 0.2 µM of unlabelled RNA and, when present, 100 nM of *E. coli* 30S subunits and 300 nM of *E. coli* initiator tRNA$^{fMet}$ (*Romilly et al., 2019*). The reaction buffer contained 10 mM Tris-acetate pH 7.6, 100 mM K$^+$-acetate, 1 mM DTT, and 10 mM Mg$^{2+}$-acetate. The RNA was denatured for 1 min at 90 °C in the presence of 5'-end-labelled primer A (3 µL, about 50,000 cpm/µL) and 0.5 mM dNTPs. After 1 min on ice, Mg$^{2+}$-acetate was added. RNAs were refolded for 5 min, followed by addition of activated 30S (10 min at 30 °C). Next, initiator tRNA$^{fMet}$ was added, and 30S-initiation complexes were allowed to form for 20 min. In the unfolded RNA sample, the 30S subunits and the initiator tRNA$^{fMet}$ were added immediately after the 1 min incubation at 90 °C. Reverse transcription was started by addition of 50 U of SuperScript IV for 20 min and stopped by adding 40 µL of cold 0.3 M Na$^+$-acetate. The reverse transcription reactions were treated as above and resolved on a 12% denaturing polyacrylamide gel.

## Homologous sequence search and alignments

Three complementary approaches were used to identify homologous sequences of *Caulobacter*'s 5'UTR$_{dnaA}$ and N$_{t\ dnaA}$. In the first one, an input nucleotide sequence including both the 5'UTR and the first 25 codons of DnaA was used in a BLASTn search against the NCBI nucleotide collection (*Altschul et al., 1990*) and the genome database (March 2020), using an E-value threshold of 10$^{-6}$. In the second approach, aiming to identify more distant homologs, the *Caulobacter* DnaA protein sequence (CCNA_00008) was used in a BLASTp search against the annotated proteins in NCBI's representative bacterial genome database (E-value cut-off 10$^{-6}$). The resulting hits were aligned with mafft 7.427 (*Katoh and Standley, 2013*), and a maximum likelihood phylogenetic tree built with IQTREE 1.6.10 (*Nguyen et al., 2015*) in order to select the cluster of the most closely related sequences for further analysis. Finally, a tBLASTn search was performed against the NCBI database of representative bacterial genomes to identify additional unannotated homologs. As previously, a tree was inferred on the hits and only sequences forming a cluster of highly related sequences were analysed further. For each hit resulting from the BLASTp and the tBLASTn searches, the upstream region containing the 5'UTR was extracted from the corresponding genome using the coordinates of the BLAST results. The results of the three homology searches were manually screened, pooled, and aligned using the MUSCLE alignment tool (*Edgar, 2004*) as implemented in SeaView 5.0.4 (*Gouy et al., 2010*).

## Computational analysis

mFold (*Zuker, 2003*) and the RNAfold server of ViennaRNA (*Gruber et al., 2008*) were used for RNA secondary structure predictions at 30 °C. For a search of putative SD sequences in the 5'UTR of *dnaA* mRNA, we used the RNAcofold tool of the ViennaRNA package to calculate the ΔG of interaction at 30 °C between an 8 nt sliding window of the 5'UTR and the anti-SD sequence of *Caulobacter*'s

16S rRNA. Three different anti-SD sequences were considered: 5'-CCUCC-3', 5'-CCUCCU-3', and 5'-CACCUCCU-3'. The structural analysis based on co-variance was performed using CMfinder 0.4.1.4 (*Yao et al., 2006*) according to the developer's indications.

## Reporter assays

A Tecan Spark multimode microplate reader was used for culture incubations as well as $OD_{600}$ and eGFP fluorescence measurements. Reporter strains were grown in 150 µL of medium at 30 °C in sterile 96-well transparent plates with flat-bottom and lid (Greiner Bio-One # 655182). Initial culture $OD_{600}$ was 0.02 (1 cm optical path length). The plates were orbitally shaken at 180 rpm with a 3 mm amplitude using a humidity cassette to protect from evaporation. $OD_{600}$ and fluorescence were measured every 15 min. The following instrument settings were used to measure eGFP fluorescence: bottom read, excitation wavelength = 485 nm (20 nm bandwidth), emission wavelength = 535 nm (25 nm bandwidth), detector gain = 100 (manual), and automatic z-position optimisation.

Raw data were processed similarly to *Zaslaver et al., 2006* using R and the splines package. The $OD_{600}$ measurements of the cultures were background-subtracted using the average value of the blank wells at each time point. To estimate the value of the background fluorescence, we grew a strain carrying the empty pMR10-BG plasmid in each plate. By plotting fluorescence against $OD_{600}$, we noticed that background changed non-linearly over the growth curve. To account for this variation, we fitted the pMR10-BG fluorescence and $OD_{600}$ data with a smoothing spline ('smooth.spline'), generating a mathematical function that allows calculation of the expected background fluorescence at any given $OD_{600}$. Background correction was performed by subtracting the expected background fluorescence to the reporter strain's fluorescence at each time point. Finally, the resulting value was normalised by dividing it with the $OD_{600}$.

$\Delta t$ and $\Delta f$ were calculated as indicated in *Figure 2*. The times of maximum fluorescence ($t_{maxFluo}$) and maximum $OD_{600}$ ($t_{maxOD}$), as well as the fluorescence values at 24 hr ($f_{24h}$) and at maximum $OD_{600}$ ($f_{maxOD}$) were calculated after fitting growth and fluorescence curves with smoothing splines ('smooth.spline'). To determine the fluorescence intensity at $OD_{600}$ = 0.05, we first determined the time at which $OD_{600}$ equals 0.05 using 'interSpline' and 'backSpline' functions. Then, the obtained value was used to calculate the corresponding fluorescence with the R function 'predict'. All statistical analyses and graph preparations were performed in GraphPad Prism (version 7).

## Translation shut-off assays

For the western blot-based translation shut-off assay, cells were cultured in 25 mL of $M2G_{1/10}$ medium for 7 hr until growth arrested. Translation was shut off by addition of chloramphenicol (100 µg/mL final concentration). 1 mL aliquots were withdrawn at 15 min intervals and snap-frozen in liquid nitrogen before being analysed by western blotting.

For the eGFP translation shut-off assay in the plate reader, reporter strains were grown on $M2G_{1/10}$ for 8.5 hr or in $M9G_{1/10}$ for 8 hr as above. When appropriate, translation was shut off by adding 100 µg/mL (*Caulobacter*) or 200 µg/mL (*E. coli*) of chloramphenicol to the culture. Fluorescence and $OD_{600}$ were measured at 10 min intervals for 10 hr. Plate reader data processing was performed as described above.

## Analysis of ribosome profiling and RNA-seq datasets

The ribosome profiling dataset in PYE (SRR1167751) and the two RNA-seq datasets (SRR1167748 and SRR1167749, M2G and PYE, respectively) were published by *Schrader et al., 2014*. The ribosome profiling datasets in M2G (SRR8570150, SRR8570151, SRR8570152) were obtained from *Aretakis et al., 2019*. Raw reads were mapped against the *C. crescentus* NA1000 NC_011916.1 reference genome using bowtie 1.2.2 (*Langmead et al., 2009*). In order to visualise mapped reads on IGV (*Robinson et al., 2011*), the sam files were converted into sorted bam files using SAMtools (*Li et al., 2009*). Coverage plots for RNA-seq and ribosome profiling were generated using the autoscale function of IGV.

## Acknowledgements

We thank members of the Jonas lab for helpful discussions and comments on the manuscript and Dr. Iker Irisarri for bioinformatic advice. The study was financially supported by the European Union's

Horizon 2020 research and innovation programme under the Marie Skłodowska-Curie grant agreement no. 797801, the Swedish Foundation for Strategic Research (FFL15-0005), the Swedish Research Council (KJ: 2016-03300, 2020-03545, EGHW: 2017-03765), and funding from the Strategic Research Area (SFO) programme distributed through Stockholm University.

## Additional information

### Funding

| Funder | Grant reference number | Author |
|---|---|---|
| H2020 Marie Skłodowska-Curie Actions | 797801 | Michele Felletti |
| Vetenskapsrådet | 2016-03300 | Kristina Jonas |
| Vetenskapsrådet | 2017-03765 | E Gerhart H Wagner |
| Swedish Foundation for Strategic Research | FFL15-0005 | Kristina Jonas |
| Stockholm University, Strategic Research Area | | Kristina Jonas |
| Vetenskapsrådet | 2020-03545 | Kristina Jonas |

The funders had no role in study design, data collection and interpretation, or the decision to submit the work for publication.

### Author contributions

Michele Felletti, Conceptualization, Data curation, Formal analysis, Funding acquisition, Investigation, Methodology, Resources, Validation, Visualization, Writing - original draft, Writing – review and editing; Cédric Romilly, Formal analysis, Investigation, Methodology, Resources, Writing – review and editing; E Gerhart H Wagner, Formal analysis, Funding acquisition, Investigation, Methodology, Resources, Writing – review and editing; Kristina Jonas, Conceptualization, Formal analysis, Funding acquisition, Investigation, Methodology, Project administration, Resources, Supervision, Validation, Writing - original draft, Writing – review and editing

### Author ORCIDs

Michele Felletti http://orcid.org/0000-0002-2494-1345
Cédric Romilly http://orcid.org/0000-0003-2129-6667
E Gerhart H Wagner http://orcid.org/0000-0003-2771-0486
Kristina Jonas http://orcid.org/0000-0002-1469-4424

### Decision letter and Author response
Decision letter https://doi.org/10.7554/eLife.71611.sa1
Author response https://doi.org/10.7554/eLife.71611.sa2

## Additional files

### Supplementary files
- Supplementary file 1. pMR10 and pZE12 reporter plasmids construction.
- Supplementary file 2. List of bacterial strains generated in this study.
- Transparent reporting form

### Data availability
All data generated or analysed during this study are included in the manuscript and supporting files. When appropriate, the raw uncropped gels and blots, the band intensity quantifications as well as the raw outputs of the computational analysis have been uploaded as "Source data" along with the figures.

The following previously published datasets were used:

| Author(s) | Year | Dataset title | Dataset URL | Database and Identifier |
|---|---|---|---|---|
| Schrader JM | 2019 | Absolute measurements of mRNA translation in *C. crescentus* reveal important fitness costs of Vitamin B12 scavenging | https://www.ncbi. nlm.nih.gov/geo/ query/acc.cgi?acc= GSE126485 | NCBI Gene Expression Omnibus, GSE126485 |
| Schrader JM, Li G, Weissman JS, Shapiro L | 2014 | The coding and noncoding architecture of the *Caulobacter crescentus* genome | https://www.ncbi. nlm.nih.gov/geo/ query/acc.cgi?acc= GSE54883 | NCBI Gene Expression Omnibus, GSE54883 |

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
