## [Decision Letter]

[Editors' note: this paper was reviewed by Review Commons.]

**Acceptance summary:**

The authors show that translation of the bacterial replication initiation protein, DnaA, is regulated in response to nutrient availability in *Caulobacter crescentus*. Regulation requires a short amino acid sequence near the N-terminus of DnaA, suggesting that the nascent protein chain can stall ribosomes in a condition-specific manner. This novel regulatory mechanism also occurs in *Escherichia coli*, and is likely to be a widespread strategy for modulating protein production in response to changes in growth conditions.

---

## [Author Response]

We thank the two reviewers for the positive assessment of our work and for the constructive comments that helped us to improve the quality of our manuscript. We have carefully considered each point and have addressed most by modifying the manuscript text to increase clarity of our work. Based on a suggestion by Reviewer 2 we have also included the results of a new experiment.

In addition to addressing all comments of the reviewers, we have expanded the part of the study analysing the functionality of *Caulobacter*’s DnaA N_t_ in the heterologous host *E. coli*. Furthermore, we have replaced our original set of fluorescence data by a new data set that has been acquired using optimized measurement parameters (bottom read and 100 for the detector gain – see Material and Methods for details), which have improved the signal-to-noise ratio and the overall quality of the fluorescence profiles. Importantly, these new data do not change, but rather strengthen, our conclusions.

Reviewer #1 (Evidence, reproducibility and clarity (Required)):Felletti et al. provide compelling new evidence that a CDS element in the dnaA mRNA is required for nutrient dependent translationol control. This provides a mechanisms by which dnaA translation is shut off during carbon starvation, and is supported by a rather rigorous analysis of the mRNA performed both in vitro and in vivo. Overall it was a pleasure to read and the data are generally very compelling. My specific comments are below:Major Comments:While the authors rule out differences in charging of different ala-tRNAs as controlling the nutrient dependent repression in translation, the authors assume that this must be due to the nascent sequence. However, could it also be possible that all ala-tRNA isoacceptors have lower charging after C-starvation?

We thank the reviewer for raising this important point. As Reviewer 1 pointed out, we cannot conclusively exclude that carbon starvation could lead to reduced charging levels of all isoacceptor Ala-tRNAs. However, based on the available literature, we consider it unlikely. In a first work by Elf et al. 2003 (confirmed later by Dittmar et al. 2005 and Subramaniam et al. 2014) the authors argued that under amino acid-limiting conditions the charging levels of the different isoacceptor tRNAs depend directly on their codon usage during translation. Importantly, in our work we could show that N_t_ mediates the inhibition of translation independent of the synonymous codon choice, suggesting that aa-tRNA levels are not limiting in our experimental conditions. To address this comment of Reviewer 1, we discussed this matter in a greater detail in the revised version of the manuscript (line 374-379).

Minor comments:It was observed many years ago that tmRNA is required for the proper timing of DNA replication initiation in Caulobacter (Cheng and Keiler J Bact 2009). Since the AAI motif is appearing to alter translation elongation, it might be interesting to discuss the AAI motif may be linked to ribosome arrest and rescue.

We appreciate this suggestion. Cheng and Keiler 2009 proposed an indirect involvement of the tmRNA in the transcriptional regulation of DnaA over *Caulobacter*’s cell cycle. In the revised version of the manuscript, we mention the tmRNA and ArfB protein as possible factors involved in ribosome rescue following Nt-induced ribosome stalling and we refer to Keiler et al. 2000 and Feaga et al. 2014.

Line 49 – add "initiation"

The word “initiation” was added to the text.

Line 61 – is "cleared" meant to be proteolyzed or simply meaning to have a lower protein level?

We apologize if we were not clear. We rephrased the text as follows: “[…] DnaA levels decrease at the onset of carbon starvation […]”.

Line 92-93 – is this 5' UTR based on a previously defined TSS determined in their previous study?

*dnaA* TSS has been first determined by primer extension (Zweiger and Shapiro 1994) and later by global 5’RACE (Schrader et al. 2014 and Zhou et al. 2015). In the new version of the manuscript, we include references to these previous studies (line 94).

Line 115-118 – this is interesting, might this conserved 5' UTR be added to rfam?

We thank the reviewer for this suggestion. We will submit our alignment to rfam after publication of the manuscript in a journal.

Line 126-127, 131,189 – Is the 3nt sequence the authors found here considered a Shine-Dalgarno site? I would imagine that this would be too small to consider this. Perhaps calling it SD-like sequence might be more appropriate.

We agree with this comment. In the new version of the manuscript, we refer to the identified 3-nucleotide sequence as a “SD-like sequence”.

Lines 136-140, 208-210 – Would the authors consider this upstream site with a potential CUG start codon a standby site? It appears to fit many of the criteria which could be used to define one.

According to our probing data, the mRNA region in proximity of the CUG start codon forms a very stable stem-loop structure. Based on our previous experience (especially the extensive work by the Wagner lab), typical ribosome standby sites only occur in largely unstructured regions. Furthermore, in Supplementary Figure 4 we show that the deletion of stem P4 does not affect eGFP expression levels. For these reasons, we consider it unlikely that the putative CUG start codon is part of a ribosome standby site.

Lines 253-255 – this is a beautiful experiment, but very hard to understand from the text. Perhaps add a sentence or two to explain it in more detail.

We thank the reviewer for this comment. In the revised version of the manuscript, we provide a more detailed description of the dfsN_t_ reporter mutant. We hope this will address the reviewer’s concerns.

Line 307 – add "synonomous"

The word “synonymous” was added in the revised version of the manuscript

When dnaA is depleted, it was observed that the chromsome can be erroneously segregated by the ParA/B/S system (mera et al. PNAS). Does this occur in C-starvation when DnaA levels drop?

In a separate study we have also observed that in a fraction of DnaA depleted cells the origin of replication is erroneously translocated from the stalked to the swarmer cell pole. We have not studied this phenomenon under carbon starvation, as it lies outside the scope of this paper. However, if the ParA/B/S remains functional under carbon starvation, this might also happen in G1-arrested starved cells.

Reviewer #1 (Significance (Required)):Appears to be quite significant to researchers studying regulation of bacterial cell cycle and translation. Since DnaA is conserved across bacteria, and this mechanism works in *E. coli*, it appears that the findings will likely be important in many bacterial systems.Referee Cross-commentingAll the reviewer comments I read seem reasonable. Specifically, I found the point about *E. coli* 30S ribosomes is very important that the authors address. This could be done in writing, but should be better listed as a caveat to those experiments.

As suggested by the reviewers, we have partially rephrased some parts of the text describing the toeprint results. Moreover, we have inserted in the main text and in Figure 1 legend explicit references to the use of purified *E. coli* 30S subunits and tRNA-fMet. We believe these changes will address the reviewers’ concerns.

Reviewer #2 (Evidence, reproducibility and clarity (Required)):Summary:The Jonas lab provide good evidence that they have found a new mechanism to regulate the amount of the DnaA protein by a starvation signal. The DnaA protein is the key chromosome replication initiator probably for most bacteria and as such DnaA is the target of many regulatory inputs. The authors created an accurate reporter system that allows them to dissect the 5' mRNA translated and untranslated sequences of dnaA and they have convincingly demonstrated that the N-terminal DnaA peptide sequence and not the RNA mediate the response to starvation by glucose exhaustion. This is potentially a model example for global translational responses in bacteria.Major comments:The main conclusion, i.e. that the DnaA leader peptide "Nt" mediates this response is convincing. However, there were 2 major problems that should be easily addressed. These do not subtract from the main conclusion.1) *E. coli* 30S subunits were used in the "Toeprint" assay of Figure 1. Obviously Caulobacter 30S Ribosome subunits should have been used, or a justification should be given. One remedy would be to make this supplementary information.

We thank Reviewer 2 for this comment. We agree that it would be better to use *Caulobacter* 30S ribosome subunits in our toeprint experiments. However, because toeprint assays with *E. coli* 30S ribosome were already established in our lab (i.e. the Wagner lab, where the assays were performed) and because works by other groups have shown that *E. coli* 30S subunits can be used to study the translation of mRNAs from other bacteria, we decided to use this experimental set up. Based on our results, we also had no reason to doubt the suitability of the *E. coli* 30S subunits. The toeprint showed that translation starts at the *in silico* predicted translation start site, which was further confirmed by our in vivo mutagenesis experiments. For these reasons, we are confident that the toeprint assays indicate the true translational start site. However, we acknowledge that we could have been more explicit about the use of the purified *E. coli* 30S subunits and tRNA-fMet in toeprinting assay. To increase clarity and transparency, in this revised version of the manuscript, some parts of the main text were rephrased and references to the use of *E. coli* 30S and tRNA-fMet were introduced (including Figure 1 legend). We hope that these changes will address the reviewer’s concerns.

2) The results in Figure 6B could be due to the Nt simply making the hybrid protein more unstable in *E. coli*. This is the main impression given by the drop in signal. In this case, the conclusion would be wrong, and Nt is not transferring a starvation translation block from C. crescentus to *E. coli*. Nt is just making the protein unstable. These results should be treated as preliminary pending protein stability measurements. However, this defect does not subtract from the other main points and without the Figure 6 *E. coli* experiments they still make a complete and interesting story. One remedy would be to make this also supplementary information.

It is indeed striking that a drop of normalised fluorescence is observed for the 5’UTR*_dnaA_*-N_t_ construct in *E. coli* but not in *Caulobacter*. In order to address if this behavior can be explained by reduced protein stability, we have performed a translation shut-off assay using the 5’UTR*_dnaA_*-N_t_
*E. coli* reporter construct. The results of this experiment (shown in Supplementary Figure 9A and described in line 327-329) show that the normalised fluorescence remains stable over 10 hours after chloramphenicol addition to the culture, ruling out that the presence of N_t_ significantly affects eGFP protein stability in *E. coli*. Importantly, this experiment also showed that in contrast to the chloramphenicol treated culture, in which the OD_600_ decreased after reaching stationary phase, the OD_600_ of the non-treated cultures slightly increased between 2 and 10 hours (Supplementary Figure 9A). Because this increase was not observed in carbon starved *Caulobacter* cultures, we consider the different growth dynamics between *E. coli* and *Caulobacter* to be the most likely explanation for differences in eGFP accumulation at later time points during the experiment.

To further strengthen our *E. coli* data, we have analysed additional relevant N_t_ mutants that we identified as most critical mutants in our *Caulobacter* experiments presented in Figure 5, namely dfsN_t_, mutD1, mutD2, ΔAAI and AAI→DDK. Determination of Δt and Δf values for the *E. coli* strains carrying these different N_t_ constructs showed similar results as for the corresponding constructs in *Caulobacter*. Collectively, these new data further support the notion that N_t_ operates in *E. coli* through a conserved inhibitory mechanism of translation. These data are now included in a reorganized new version of Figure 6 (panels A, B) as well as in Supplementary Figure 9.

Minor comments:There are also 6 minor issues that are easily addressed, most by small changes to the text, and these should improve this otherwise fine manuscript.1) Line 88 Figure 1A shows DnaA degradation upon entering stationary phase from a low glucose media and not a simple starvation response to one component like glucose. Did the authors consider trying simple washout experiments, i.e. resuspend the cells in glucose-free media? This would have the advantage of suddenly exposing the cells to starvation and thereby studying the sudden response rather than the slower lingering response which would be due to many factors and not just glucose removal.

In a previous work from our lab (Leslie et al. 2015), we have conclusively shown that the downregulation of DnaA synthesis depends primarily on the nutrient content of the growth medium.

Besides being in continuity with our previous work, we think that the starvation protocol that we used in the present study, and that was also used by the Sean Crosson lab (Boutte et al. 2012), might better reproduce what happens in the natural environment when nutrient levels gradually decrease until becoming limiting for bacterial growth.

2) Reference 16 should be cited are the first publication to show that glucose and other starvations induce DnaA degradation in Caulobacter.

We have added Reference 16 to the first sentence of the Results section, in which we state that DnaA levels decrease when cells are shifted from a glucose-supplemented minimal medium to a glucose-limiting medium.

3) Figure 1D shows that the TOEprint is not changed by adding the ribosome, very surprising considering its size and SD docking and alignment. 2 Minor bands then appear when the tRNA-Met is further added. These are presumably the "toeprints". A control with just the added tRNA-Met would make this result much more significant.

In the existing literature, there is a common consensus to consider real toeprints (i.e., indicative of the presence of an assembled translation pre-initiation complex) as only those bands that appear faintly in the presence of the 30S ribosome subunit but that become clearly enhanced upon addition of the initiator tRNA-fMet. Some examples can be found in Hoekzema et al. 2019, Romilly 2014, Romilly 2020. In cases when the translation start site is buried in a structural element, the intensity of the toeprint signal is further increased when the mRNA is rendered unfolded, as seen in our data.

tRNA-30S-independent bands always show up in toeprint experiments, but their intensities differ with the sequence of the mRNA and sometimes the choice of RT used for primer extension. Addition of initiator tRNA-fMet alone is commonly not done in toeprint experiments (see references mentioned above). Finally, we want to point out again (see also our answer on “Problem 1”) that the toeprint data are very much consistent with our *in silico* predictions and our in vivo mutagenesis data. Therefore, we are confident that the observed toeprint upstream of the AUG corresponds to the true ribosome binding site.

4) Why does the cell OD drop, e.g. in Figure 2, is it cell death from starvation?

We don’t think that the slight reduction of OD_600_ observed in our experiments is due to cell death. Based on our knowledge, carbon starved cells remain viable up to 24 hours after the starvation onset. Instead, we have observed a cell volume reduction, which may at least partially explain the observed OD_600_ decrease.

5) Line 327 Discussion "This study reveals a new mechanism, by which some bacteria can regulate the synthesis of the replication initiator DnaA in response to nutrient availability by modulating the rate of translation." Rate of translation or rate of translation abortions (as implied in Figure 6)?

The rate of translation is the result of multiple contributions such as initiation, elongation, abortion and termination. Our data indicate that N_t_ is a regulator of DnaA translation elongation responding specifically to the nutritional state of the cell. Translation abortion could be one of the possible outcomes (but not the only one) of ribosome stalling. For these reasons, in the new version of the manuscript, we added the word “elongation” at the end of the sentence mentioned by Reviewer 2 (line 354).

6) It seems that for most experiments with the eGFP the translation and protein decay components of the signal could have been easily uncoupled by running a parallel +chloramphenicol control. For example, this would simplify the interpretation of Figure 6 where Nt eGFP stabilities are an issue and it is important to establish that comparable protein stability with and without the Nt peptide.

To address the reviewer’s comment, we have now included a chloramphenicol control experiment (stability assay) performed with *E. coli* carrying the 5’UTR*_dnaA_*-N_t_ reporter construct (Supplementary Figure 9A). Please, see the response above for more details. For the experiments with the *Caulobacter* 5’UTR*_dnaA_*-N_t_ reporter we show in Supplementary Figure 7 that the N_t_ peptide has no destabilising effect on eGFP.

Reviewer #2 (Significance (Required)):*Caulobacter crescentus* is a model bacterium that has provided many insights into bacterial physiology that are now exploited to understand many organisms. These present results may provide one such example. It is known that the first amino acids of translated peptides can influence increase or impede exit from the ribosome, so this is a potential translation-level regulatory point that might be used by many organisms. This manuscript gives a concrete and important example of such usage suggesting that it many be widespread. Therefore, this work should find a wide audience and it should stimulate research in many other systems.My lab also studies *Caulobacter crescentus* and we studied the same dnaA gene and protein including starvation responses. We at present do not have projects on dnaA but we do study other regulators and regulatory mechanisms of chromosome replication in *Caulobacter crescentus*.